# Probabilistic and Differentiable Wireless Simulation with Geometric Transformers

## Abstract

Modelling the propagation of electromagnetic signals is critical for designing modern communication systems. While there are precise simulators based on ray tracing, they do not lend themselves to solving inverse problems or the integration in an automated design loop. We propose to address these challenges through differentiable neural surrogates that exploit the geometric aspects of the problem. We first introduce the Wireless Geometric Algebra Transformer (Wi-GATr), a generic backbone architecture for simulating wireless propagation in a 3D environment. It uses versatile representations based on geometric algebra and is equivariant with respect to $E(3)$, the symmetry group of the underlying physics. Second, we study two algorithmic approaches to signal prediction and inverse problems based on differentiable predictive modelling and diffusion models. We show how these let us predict received power, localize transmitters, and reconstruct the 3D environment from the received signal. Finally, we introduce two large, geometry-focused datasets of wireless signal propagation in indoor scenes. In experiments, we show that our geometry-forward approach achieves higher-fidelity predictions with less data than various baselines.

## 1 Introduction

Modern communication is wireless: more and more, we communicate via electromagnetic waves through the antennas of various devices, leading to progress in and adoption of mobile phones, automotive, AR/VR, and IoT technologies [12, 16]. All these innovations build upon electromagnetic wave propagation. Therefore, modelling and understanding wave propagation in space is a core research area in wireless communication, and remains crucial as we are moving toward new generations of more efficient and spatially-aware wireless technologies.

Wireless signal propagation follows Maxwell's equations of electromagnetism and is often accurately modelled by state-of-the-art ray-tracing simulation software. However, these simulators take substantial time to evaluate for each scene, cannot be fine-tuned on measurements, and are (usually [29]) not differentiable. This limits their usefulness for solving inverse problems.

In contrast, neural models of signal propagation can be evaluated cheaply, can be trained on real measurements in addition to simulation, and are differentiable and thus well-suited for solving inverse problems. Several such approaches have been proposed recently, often using image-based representations of the inputs and outputs and off-the-shelf vision architectures [6, 23, 34, 35, 44, 46, 51, 52]. However, wireless surrogate modelling faces various challenges. Realistic training data is often scarce, requiring surrogate models to be data efficient. Wireless environments can consist of complex meshes. Finally, input and output data consist of a variety of data types, including the shape of extended 3D objects, point coordinates and spatial orientation of antennas, and information associated with the transmitted signal.

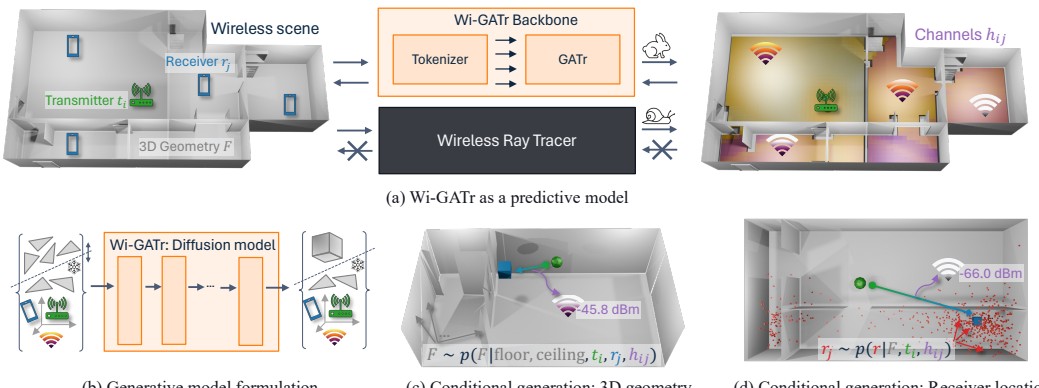

Figure 1: **Geometric surrogates for modelling wireless signal propagation.** **(a)**: Predictive modelling of channels from 3D geometry, transmitter, and receiver properties. Wi-GATr is a fast and differentiable surrogate for ray tracers. **(b)**: A probabilistic approach with diffusion models lets us reconstruct 3D environments **(c)** and antenna positions **(d)** from the wireless signal.

In this work, we present a new approach to modelling wireless signal propagation. It is grounded in the observation that wireless propagation is inherently a geometric problem: a directional signal is transmitted by an oriented transmitting antenna, the signal interacts with surfaces in the environment, and the signal eventually impinges an oriented receiving antenna. We argue that it is critical for neural surrogates to model and flexibly represent geometric aspects (e. g. orientations, shapes) in the propagation environment. We therefore develop surrogate models based on flexible geometric representation and strong geometric inductive biases.

We first propose the *Wireless Geometric Algebra Transformer* (Wi-GATr), a backbone architecture for wireless signal propagation problems. A key component is a new tokenizer for the diverse, geometric data of wireless scenes. The tokens are processed with a Geometric Algebra Transformer (GATr) network [9]. This architecture is equivariant with respect to the symmetries of wireless channel modelling, but maintains the scalability of a transformer architecture.

Second, we study Wi-GATr models as differentiable, predictive surrogates for the simulator (see Fig. 1a). Here the network predicts observables such as the received power as a function of transmitter position, receiver position, and 3D environment. We show how this enables forward modelling, and in addition, inverse problem solving due to Wi-GATr's differentiability.

Next, we propose an alternative, more versatile probabilistic approach to prediction and inference tasks: training Wi-GATr diffusion models (Fig. 1b) on the joint distribution of transmitter, receiver, channel information, and 3D environment. At test time, the model can be flexibly conditioned on any available information to predict the received power, localize a transmitter or receiver (Fig. 1c), or even reconstruct the (full or partial) 3D geometry from the wireless signal (Fig. 1d).

To enable machine learning development for wireless problems, we finally introduce two new datasets, `Wi3R` and `WiPTR`. Each dataset consists of thousands of indoor scenes of varying complexity and include all the geometric information that characterizes a wireless scene.

Finally, we demonstrate the predictive and the probabilistic models on these datasets. Our experiments show that the Wi-GATr approach gives us a higher-fidelity predictions than various baselines, generalizes robustly to unseen settings, and requires up to 20 times less data for the same performance than a transformer baseline.

## 2 Background and related work

**Wireless signal propagation.** How do wireless signals propagate from a transmitting antenna (Tx) to a receiver antenna (Rx) in a (static) 3D environment? While the system is fundamentally described by Maxwell's equations, for many realistic problems the ray approximation of geometric optics suffices [31]. It approximates the solution to Maxwell's equations as a sum of planar waves propagating in all directions from Tx. Each planar wave is represented as a ray, characterized by various attributes (e. g., power, phase, delay) since transmission. As a ray reaches an object—that is,

it intersects with its mesh—the interaction is modelled as reflection, refraction, or diffraction. During such interactions, the power, phase, polarization, and propagation direction of the wave can change in complex, material-dependent ways. In addition, new rays can emanate from the point of interaction. After multiple interactions, the rays eventually reach the receiving antenna. The Tx and Rx are then linked by a connected path $p$ of multiple rays. The effects on the received signal are described by the channel impulse response (CIR) $h(\tau) = \sum_p a_p \delta(\tau - \tau_p)$, where $a_p \in \mathbb{C}$ is the complex gain and $\tau_p$ the delay of the incoming rays [53].

Maxwell's equations and in extension ray propagation are highly symmetric. The received signal does not change under rotations, translations, and reflections of the whole scene, as well as the exchange of transmitter and receiver. The latter property is known as reciprocity [37].

**Wireless simulators.** Wireless propagation models play a key role in design and evaluation of communication systems, for instance by characterizing the gain of competitive designs in *realistic* settings or by optimizing systems performance as in base station placement for maximal coverage. Statistical approaches [2] represent propagation as a generative model where the parameters of a probabilistic model are fitted to measurements. On the other hand, wireless ray-tracing approaches [1, 5, 29] are increasingly popular due to their high accuracy and because they do not require expensive field measurement collection campaigns.

**Neural wireless simulations.** Both statistical and ray-tracing simulation techniques are accompanied by their own shortcomings, subsequently mitigated by their neural counterparts. Neural surrogates for statistical models [19, 40, 42, 56] reduce the amount and cost of measurements required. Neural ray tracers [29, 41, 58] address the non-differentiability of simulators using a NeRF-like strategy [38] by parameterizing the scene using a spatial MLP and rendering wireless signals using classic ray-tracing or volumetric techniques. While these techniques are faster than professional ray tracers, they are similarly bottlenecked by expensive bookkeeping and rendering steps (involving thousands of forward passes). In contrast, we propose a framework to simulate wireless signals with a single forward pass through a geometric transformer that is both sample-efficient and generalizes to novel scenes.

**Geometric deep learning.** The growing field of geometric deep learning [11] aims to incorporate structural properties of a problem into neural network architectures and algorithms. A central concept is *equivariance* to symmetry groups [15]: a network $f(x)$ is equivariant with respect to a group $G$ if its outputs transform consistently with any symmetry transformation $g \in G$ of the inputs, $f(g \cdot x) = g \cdot f(x)$, where $\cdot$ denotes the group action. Of particular interest to us is the Euclidean group $E(3)$ of isometries of 3D space, that is, transformations that leave Euclidean distances invariant. This group includes spatial translations, rotations, reflections, and their combinations. As we argued above, the physics of wireless signal propagation are invariant under this group.

**GATr.** The Geometric Algebra Transformer (GATr) [9] is an $E(3)$-equivariant architecture for geometric problems. Among equivariant architectures, it stands out in two ways. First, it uses geometric (or Clifford) algebras [14, 22] as representations. For a rigorous introduction to these algebras, we refer the reader to Dorst [20]. From a practical machine learning perspective, these algebras define embeddings for various geometric primitices like 3D points, planes, or $E(3)$ transformations. We will show that this representation is particularly well-suited for wireless channel modelling. Second, GATr is a transformer architecture [54]. It computes the interactions between multiple tokens through scaled dot-product attention. With efficient backends like FlashAttention [17], the architecture is scalable to large systems, without any restrictions on the sparsity of interactions like in message-passing networks.

**Diffusion models.** Diffusion models [25, 48, 50] are a class of generative models that iteratively invert a noising process. They have become the de-facto standard in image and video generation [26, 45]. Recently, they have also shown to yield promising results in the generation of spatial and sequential data, such as in planning [30] and puzzle solving [28]. Aside from their generative modelling capabilities, diffusion models provide a flexible way for solving inverse problems [13, 36] through multiplication with an appropriate likelihood term [48]. Furthermore, by combining an invariant prior distribution with an equivariant denoising network, one obtains equivariant diffusion models [33]. These yield a sampling distribution that assigns equal probability to all symmetry transformations of an object, which can improve performance and data efficiency in symmetry problems like molecule generation [27] and planning [10]. We will demonstrate similar benefits in modelling wireless signal propagation.

## 3 The Wireless Geometric Algebra Transformer (Wi-GATr)

### 3.1 Problem formulation

Our goal is to model the interplay between 3D environments, transmitting and receiving antennas, and the resulting transmitted wireless signals. More precisely, we consider *wireless scenes* consisting of:

- The 3D geometry $F$ of the environment. We specify it through a triangular mesh with a discrete material class associated with each mesh face.
- A set of transmitting antennas $t_i$ for $i = 1, \ldots, n_t$. Each $t_i$ is characterized by a 3D position, an orientation, and any antenna characteristics. We will often focus on the case of a single Tx and then omit the index $i$.
- Analogously, a set of receiving antennas $r_i$ for $i = 1, \ldots, n_r$.
- The channel or signal $h_{ij}$ between each transmitter $i$ and each receiver $j$, which can be any observable function of the CIR.

In this setting, we consider various downstream tasks:

- *Signal prediction* is about predicting the signal received at a single antenna from a single receiver, $p(h|F, t, r)$ with $n_t = n_r = 1$. This is exactly the task that ray-tracing simulators solve. Often, the signal is modelled deterministically as a function $h(F, t, r)$.
- *Receiver localization*: inferring the position and properties of a receiving antenna from one or multiple transmitters, $r \sim p(r|F, \{t_i\}, \{h_i\})$, with $n_r = 1$.
- *Geometry reconstruction* or sensing: reconstructing a 3D environment partially, inferring $p(F_u|F_k, t, r, h)$, where $F_u$ and $F_k$ are the unknown and known subsets of $F$, respectively.

The latter two problems are examples of *inverse problems*, as they invert the graphical model that simulators are designed for. They are not straightforward to solve with the simulators directly, but we will show how neural surrogates trained on simulator data can solve them.

### 3.2 Backbone

Core to our approach to this family of inference problems is the Wireless Geometric Algebra Transformer (Wi-GATr) backbone. It consists of a novel tokenizer and a network architecture.

**Wireless GA tokenizer.** The tokenizer takes as input some subset of the information characterizing a wireless scene and outputs a sequence of tokens that can be processed by the network. A key challenge in the neural modelling of wireless problems is the diversity of types of data involved. As we argued above, a wireless scene consists of the 3D environment mesh $F$, which features three-dimensional objects such as buildings and trees, antennas $t$ and $r$ characterized through a point-like position, an antenna orientation, and additional information about the antenna type, and the characteristics of the channel $h$.

| Data type | Input parameterization | Tokenization | Channels ($\mathbb{G}_{3,0,1}$ embedding) |
| --- | --- | --- | --- |
| 3D environment $F$ | • Triangular mesh | 1 token per mesh face | • Mesh face center (point) |
| | | | • Vertices (points) |
| | | | • Mesh face plane (oriented plane) |
| | • Material classes | | • One-hot material emb. (scalars) |
| Antenna $t_i/r_i$ | • Position | 1 token per antenna | • Position (point) |
| | • Orientation | | • Orientation (direction) |
| | • Receiving/transmitting | | • One-hot type embedding (scalars) |
| | • Additional characteristics | | • Characteristics (scalars) |
| Channel $h_{ij}$ | • Antennas | 1 token per link | • Tx position (point) |
| | | | • Rx position (point) |
| | | | • Tx-Rx vector (direction) |
| | • Received power | | • Normalized power (scalar) |
| | • Phase, delay, ... | | • Additional data (scalars) |

Table 1: **Wireless GA tokenizer.** We describe how the mesh parameterizing the 3D environment and the information about antennas and their links are represented as a sequence of geometric algebra tokens. The mathematical representation of $\mathbb{G}_{3,0,1}$ primitives like points or orientated planes is described in Appendix A.

To support all of these data types, we propose a new tokenizer that outputs a sequence of geometric algebra (GA) tokens. Each token consists of a number of elements (channels) of the projective geometric algebra $\mathbb{G}_{3,0,1}$ in addition to the usual unstructured scalar channels. We define the GA precisely in Appendix A. Its main characteristics are that each element is a 16-dimensional vector and can represent various geometric primitives: 3D points including an absolute position, lines, planes, and so on. This richly structured space is ideally suited to represent the different elements encountered in a wireless problem. Our tokenization scheme is specified in Tbl. 1.

**Network.** After tokenizing, we process the input data with a Geometric Algebra Transformer (GATr) [9]. This architecture naturally operates on our $\mathbb{G}_{3,0,1}$ parameterization of the scene. It is equivariant with respect to permutations of the input tokens as well as E(3), the symmetry group of translations, rotations, and reflections. These are exactly the symmetries of wireless signal propagation, with one exception: wireless signals have an additional reciprocity symmetry that specifies that the signal is invariant under an role exchange between transmitter and receiver. We will later show how we can incentivize this additional symmetry property through data augmentation.[1] Finally, because GATr is a transformer, it can process sequences of variable lengths and scales well to systems with many tokens. Both properties are crucial for complex wireless scenes, which can in particular involve a larger number of mesh faces.

## 3.3 Predictive modelling

The Wi-GATr backbone can be used either in a predictive or probabilistic ansatz. We begin with the predictive modelling of the measured channel information as a function of the complete 3D environment and the information characterizing the transmitter and receiver, $h_\theta(F, t, r)$. This regression model is trained in a supervised way on simulated or measured wireless scenes.

**Forward prediction.** The network thus learns a differentiable, deterministic surrogate for the simulator model $h_{\text{sim}}(F, t, r)$. At test time, we can use the network instead of a simulator to predict the signals in unseen, novel scenes. Compared to a simulator based on ray tracing, it has three advantages: it can be evaluated in microseconds rather than seconds or minutes, it can be finetuned on real measurements, and it is differentiable.

**Inverse problems.** This differentiability makes such a surrogate model well-suited to solve inverse problems. For instance, we can use it for receiver localization. Given a 3D environment $F$, transmitters $\{t_i\}$, and corresponding signals $\{h_i\}$, we can find the most likely receiver position and orientation as $\hat{r} = \arg\min_r \sum_i \|h_\theta(F, t_i, r) - h\|^2$. The minimization can be performed numerically through gradient descent, thanks to the differentiability of the Wi-GATr surrogate.

## 3.4 Probabilistic modelling

While a predictive model of the signal can serve as a powerful neural simulator, it has two shortcomings. Solving an inverse problem through gradient descent requires a sizable computational cost for every problem instance. Moreover, predictive models are deterministic and do not allow us to model stochastic forward processes or express the inherent uncertainty in inverse problems.

**Equivariant diffusion model.** To overcome this, we draw inspiration from the inverse problem solving capabilities of diffusion models using guidance [13]. In this case, we formulate the learning problem as a generative modelling task of the joint distribution $p_\theta(F, t, r, h)$ between 3D environment mesh $F$, transmitter $t$, receiver $r$, and channel $h$, for a single transmitter-receiver pair. Concretely, we follow the DDPM framework and use a Wi-GATr model as score estimator (denoising network). By using an invariant base density and an equivariant denoising network, we define an invariant generative model. See Appendix B for a detailed description of our diffusion model and the discussion of some subtleties in equivariant generative modelling.

**Unifying forward prediction and inverse problems as conditional sampling.** A diffusion model trained to learn the joint density $p_\theta(F, t, r, h)$ does not only allow us to generate unconditional samples of wireless scenes, but also lets us sample from various conditionals: given a partial wireless scene, we can fill in the remaining details, in analogy to how diffusion models for images allow for

---

[1]We also experimented with a reciprocity-equivariant variation of the architecture, but that led to a marginally worse performance without a significant gain in sample efficiency.

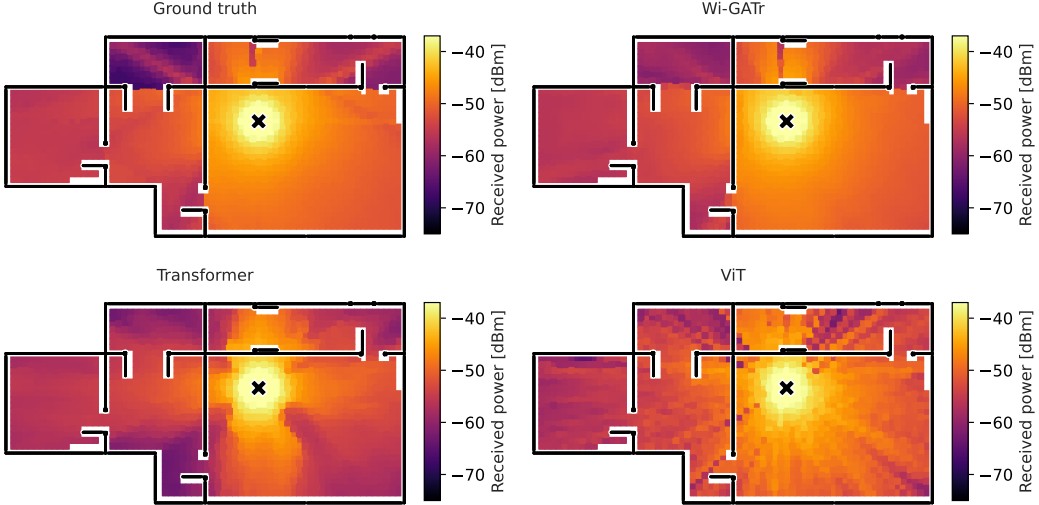

**Figure 2: Qualitative signal prediction results.** We show a single floor plan from the `WiPTR` test set. The black lines indicate the walls and doors, the colors show the received power as a function of the transmitter location (brighter colours mean a stronger signal). The transmitting antenna is shown as a black cross. The $z$ coordinates of transmitter and receiver are all fixed to the same height. We compare the ground-truth predictions (top left) to the predictions from different predictive models, each trained on only 100 `WiPTR` floor plans. Wi-GATr is able to generalize to this unseen floor plan even with such a small training set.

inpainting. To achieve this, we use the conditional sampling algorithm proposed by Sohl-Dickstein et al. [48]: at each step of the sampling loop, we fix the conditioning variables to their known values before feeding them into the denoising network.

This algorithm lets us solve signal prediction (sampling from $p_\theta(h|F,t,r)$), receiver localization (from $p_\theta(r|F,t,h)$), geometry reconstruction (from $p_\theta(F_u|F_k,t,r,h)$), or any other inference task in wireless scenes. We thus unify "forward" and "inverse" modelling in a single algorithm. Each approach is probabilistic, enabling us to model uncertainties. This is important for inverse problems, where measurements often underspecify the solutions.

In principle, the unconditional diffusion objective should suffice to enable test-time conditional sampling. In practice, we find that we can improve the conditional sampling performance with two modifications. First, we combine training on the unconditional diffusion objective with conditional diffusion objectives. For the latter, we randomly select tokens to condition on and evaluate the diffusion loss only on the remaining tokens. Second, we provide the conditioning mask as an additional input to the denoising model. See Appendix B for details.

## 4 New datasets

While several datasets of wireless simulations and measurements exist [3, 4, 41, 57], they either do not include geometric information, are not diverse, are at a small scale, or the signal predictions are not realistic. To facilitate the development of machine learning methods with a focus on geometry, we generate two new datasets of simulated wireless scenes.[2] Both feature indoor scenes and channel information generated with a state-of-the-art ray-tracing simulator [1] at a frequency of 3.5 GHz. They provide detailed characteristics for each path between Tx and Rx, such as gain, delay, angle of departure and arrival at Tx/Rx, and the electric field at the receiver itself, which allows users to compute various quantities of interest themselves. See Appendix C for more details.

`Wi3R` **dataset.** Our first dataset focuses on simplicity: each of 5000 floor plans has the same size and number of rooms, and all walls have the same material across layouts. They differ only in their layouts, which we take from `Wi3Rooms` [41], Tx positions, and Rx positions. In Appendix C we define training, validation, and test splits as well as an out-of-distribution set to test the robustness of different models.

---

[2]We are preparing the publication of the datasets.

| | Wi3R dataset | | | | WiPTR dataset | | |
|---|---|---|---|---|---|---|---|
| | Wi-GATr (ours) | Transf. | SEGNN | PLViT | Wi-GATr (ours) | Transf. | PLViT |
| *In distribution* | | | | | | | |
| Rx interpolation | **0.63** | 1.14 | 0.92 | 5.61 | **0.53** | 0.84 | 1.67 |
| Unseen floor plans | **0.74** | 1.32 | 1.02 | 5.84 | **0.54** | 0.87 | 1.66 |
| *Symmetry transformations* | | | | | | | |
| Rotation | **0.74** | 78.68 | 1.02 | 5.84 | **0.54** | 28.17 | 1.66 |
| Translation | **0.74** | 64.05 | 1.02 | 5.84 | **0.54** | 4.04 | 1.66 |
| Permutation | **0.74** | 1.32 | 1.02 | 5.84 | **0.54** | 0.87 | 1.66 |
| Reciprocity | **0.74** | 1.32 | 1.01 | 8.64 | **0.54** | 0.87 | 1.65 |
| *Out of distribution* | | | | | | | |
| OOD layout | 9.24 | 14.06 | **2.34** | 7.00 | **0.54** | 1.01 | 1.58 |

**Table 2: Signal prediction results.** We show the mean absolute error on the received power in dBm (lower is better, best in bold). **Top**: In-distribution performance. **Middle**: Generalization under symmetry transformations. **Bottom**: Generalization to out-of-distribution settings. In almost all settings, Wi-GATr is the highest-fidelity surrogate model.

`WiPTR` **dataset.** Next, we generate a more varied, realistic dataset based on the floor layouts in the `ProcTHOR-10k` dataset for embodied AI research [18]. We extract the 3D mesh information including walls, windows, doors, and door frames and assign 6 different dielectric materials for different groups of objects. Our dataset consists of 12k different floor layouts, split into training, test, validation, and OOD sets as described in Appendix C. Not only does `WiPTR` stand out among wireless datasets in terms of its level of detail and scale, but because it is based on `ProcTHOR-10k`, it is also suited for the integration with embodied AI research.

## 5 Experiments

### 5.1 Predictive modelling

We focus on the prediction of the time-averaged non-coherent received power $h = \sum_p |a_p|^2$, disregarding delay or directional information that may be available in real measurements. We train predictive surrogates $h_\theta(F, t, r)$ that predict the power as a function of the Tx position and orientation $t$, Rx position and orientation $r$, and 3D environment mesh $F$, on both the `Wi3R` and `WiPTR` datasets. All models are trained with reciprocity augmentation, i.e., randomly flipping Tx and Rx labels during training. This improves data efficiency slightly, especially for the transformer baseline.

In addition to our Wi-GATr model, described in Sec. 3, we train several baselines. The first is a vanilla transformer [54], based on the same inputs and tokenization of the wireless scene, but without the geometric inductive biases. Next, we compare to the E(3)-equivariant SEGNN [8], though we were only able to fit this model into memory for the `Wi3R` dataset. In addition, we train a PLViT model, a state-of-the-art neural surrogate for wireless scenes [24] that represent wireless scenes as an image centered around the Tx position. Finally, we attempt to compare Wi-GATr also to WiNeRT [41], a neural ray tracer. However, this architecture, which was developed to be trained on several measurements on the same floor plan, was not able to achieve useful predictions on our diverse datasets with their focus on generalization across floor plans. Our experiment setup and the baselines are described in detail in Appendix D.

**Signal prediction.** In Fig. 2 we illustrate the prediction task on a `WiPTR` floor plan. We show signal predictions for the simulator as well as for surrogate models trained on only 100 floor plans. Despite this floor plan not being part of the training set, Wi-GATr is able to capture the propagation pattern well, while the transformer and ViT show memorization artifacts.

In Tbl. 2 we compare surrogate models trained on the full `Wi3R` and `WiPTR` datasets. Both when interpolating Rx positions on the training floor plans as well as when evaluating on new scenes unseen during training, Wi-GATr offers the highest-fidelity approximation of the simulator. Wi-GATr as well as the equivariant baselines are by construction robust to symmetry transformations, while

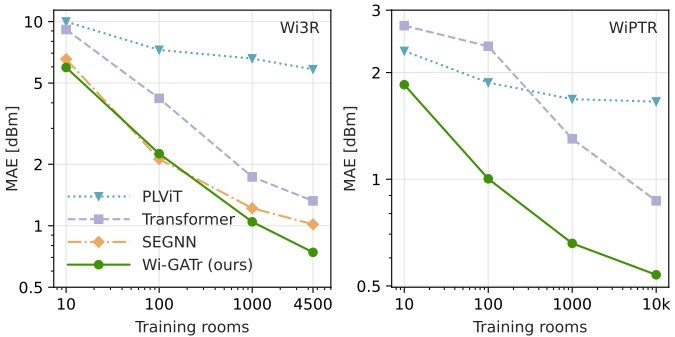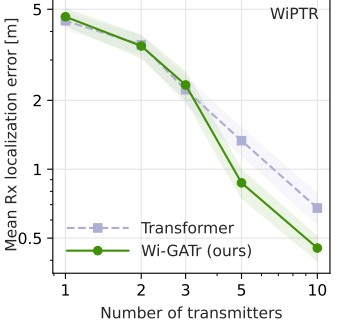

**Figure 3: Signal prediction.** We show the mean absolute error on the received power as a function of the training data on `Wi3R` (left) and `WiPTR` (right). Wi-GATr outperforms the transformer and PLViT baselines at any amount of training data, and scales better to large data or many tokens than SEGNN.

**Figure 4: Rx localization error**, as a function of the number of Tx. Lines and error band show mean and its standard error over 240 measurements.

the performance of a vanilla transformer degrades substantially. All methods but SEGNN struggle to generalize to an OOD setting on the `Wi3R` dataset. This is not surprising given that the training samples are so similar to each other. On the more diverse `WiPTR` dataset, Wi-GATr is almost perfectly robust under domain shift.

**Data efficiency.** Next, we study the data efficiency of the different surrogates in Fig. 3. Wi-GATr is more data-efficient than any other method with the exception of the $E(3)$-equivariant SEGNN, which performs similarly well for a small number of training samples. This confirms that equivariance is a useful inductive bias when data is scarce. But Wi-GATr scales better than SEGNN to larger number of samples, showing that our architecture combines the small-data advantages of strong inductive biases with the large-data advantages of a transformer architecture.

**Inference speed.** One of the advantages of neural surrogates is their test-time speed. Both Wi-GATr and a transformer are over a factor of 20 faster than the ground-truth ray tracer (see Appendix D).

**Receiver localization.** Next, we show how differentiable surrogates let us solve inverse problems, focusing on the problem of receiver localization. We infer the Rx position with the predictive surrogate models by optimizing through the neural surrogate of the simulator as discussed in Sec. 3.3. The performance of our surrogate models is shown in Fig. 4 and Appendix D.[3] The two neural surrogates achieve a similar performance when only one or two transmitters are available, a setting in which the receiver position is highly ambiguous. With more measurements, Wi-GATr lets us localize the transmitter more precisely.

## 5.2 Probabilistic modelling

Next, we experiment with our probabilistic approach. We train diffusion models on the `Wi3R` dataset. In addition to a Wi-GATr model, we study a transformer baseline, as well as a transformer trained on the same data augmented with random rotations. Both models are trained with the DDPM pipeline with 1000 denoising steps and samples from with the DDIM solver [49]. Our setup is described in detail in Appendix D.

**Signal prediction, receiver localization, and geometry reconstruction as conditional sampling.** In our probabilistic approach, signal prediction, receiver localization, and geometry reconstruction are all instances of sampling from conditional densities: $h \sim p_\theta(h|F, t, r)$, $r \sim p_\theta(r|F, t, h)$, and $F_u \sim p_\theta(F_u|F_k, t, r, h)$, respectively. We qualitatively show results for this approach in Figs. 1 and 5. All of these predictions are probabilistic, which allows our model to express uncertainty in ambiguous inference tasks. When inferring Rx positions from a single measurement, the model learns multimodal densities, as shown in the middle of Fig. 5. When reconstructing geometry, the model will sample diverse floor plans as long as they are consistent with the transmitted signal, see the right panel of Fig. 5. Additional results on signal and geometry prediction are given in Appendix D.2.

---

[3]Neither the SEGNN nor PLViT baselines are fully differentiable with respect to object positions when using the official implementations from Refs. [7, 24]. We were therefore not able to accurately infer the transmitter positions with these architectures.

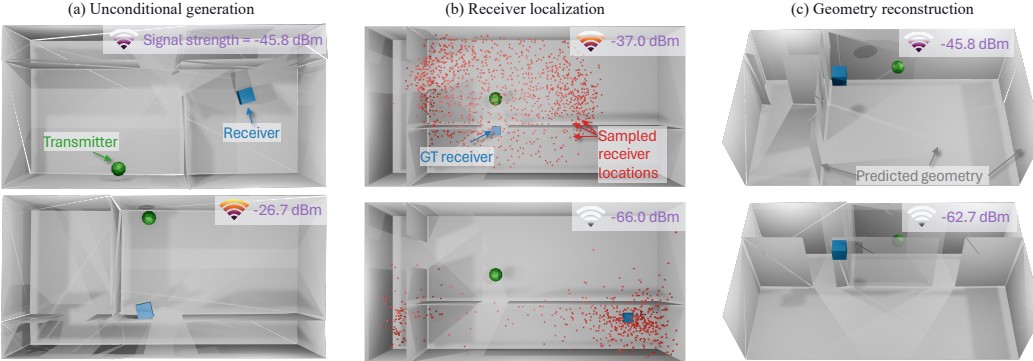

**Figure 5: Probabilistic modelling.** We formulate various tasks as sampling from the unconditional or conditional densities of a single diffusion model. **(a)**: Unconditional sampling of wireless scenes $p(F, t, r, h)$. **(b)**: Receiver localization as conditional sampling from $p(r|F, t, h)$ for two different values of $h$ and $r$. **(c)**: Geometry reconstruction as conditional sampling from $p(F_u|F_k, t, r, h)$ for two different values of $h$, keeping $t, r, F_k$ fixed.

We quantitatively evaluate these models through the variational lower bound on the log likelihood of test data under the model. To further analyze the effects of equivariance, we test the model both on canonicalized scenes, in which all walls are aligned with the $x$ and $y$ axis, and scenes that are arbitrarily rotated. The results in Tbl. 3 show that Wi-GATr outperforms the transformer baseline across all three tasks, even in the canonicalized setting or when the transformer is trained with data augmentation. The gains of Wi-GATr are particularly clear on the signal prediction and receiver localization problems.

| | Wi-GATr (ours) | Transformer | |
| --- | --- | --- | --- |
| | | default | data augm. |
| *Canonicalized scenes* | | | |
| Signal pred. | **1.62** | 3.00 | 15.66 |
| Receiver loc. | **3.64** | 8.28 | 14.42 |
| Geometry reco. | **-3.95** | -3.61 | -2.10 |
| *Scenes in arbitrary rotations* | | | |
| Signal pred. | **1.62** | 9.57 | 17.65 |
| Receiver loc. | **3.64** | 105.68 | 14.45 |
| Geometry reco. | **-3.95** | 389.34 | -2.34 |

**Table 3: Probabilistic modelling results**. We show variational upper bounds on the negative log likelihood for different conditional inference tasks (lower is better, best in bold).

## 6 Discussion

Wireless signal transmission through electromagnetic wave propagation is an inherently geometric and symmetric problem. We developed a class of neural surrogates grounded in geometric representations and strong inductive biases. They are based on our new Wi-GATr backbone architecture, consisting of a new tokenization scheme for wireless scenes together with an E(3)-equivariant transformer architecture. The proposed backbone is applied in two ways to wireless tasks: first, as a differentiable "forward" prediction model that maps the features to the signals; second, as a probabilistic diffusion model that captures the joint and conditional distributions of features and channels. We employed these designs in experiments on received power prediction, receiver localization, and geometry reconstruction, where our Wi-GATr models enabled precise predictions, outperforming various baselines.

Our analysis is in many ways a first step. The range of materials in our datasets is limited and we only experimented with measurements of the non-coherent total received power, which is a stable signal, but offers less spatial information than measurements of the time delay or angular information. More importantly, we only considered idealized inference tasks. For instance, our receiver localization problem assumed perfect knowledge of the room geometry and materials.

Nevertheless, we hope that we were able to highlight the benefits of a geometric treatment of wave propagation modelling. Augmenting or replacing the image-based or general-purpose representations and architectures prevalent in wireless modelling with geometric approaches has the potential of improving data efficiency, performance, and robustness.

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

# A  Geometric algebra

As representation, Wi-GATr uses the projective geometric algebra $\mathbb{G}_{3,0,1}$. Here we summarize key aspects of this algebra and define the canonical embedding of geometric primitives in it. For a precise definition and pedagogical introduction, we refer the reader to Dorst [20].

**Geometric algebra.**  A geometric algebra $\mathbb{G}_{p,q,r}$ consists of a vector space together with a bilinear operation, the *geometric product*, that maps two elements of the vector space to another element of the vector space.

The elements of the vector space are known as *multivectors*. Their space is constructed by extending a base vector space $\mathbb{R}^d$ to lower orders (scalars) and higher-orders (bi-vectors, tri-vectors, . . . ). The algebra combines all of these orders (or *grades*) in one $2^d$-dimensional vector space. From a basis for the base space, for instance $(e_1, e_2, e_3)$, one can construct a basis for the multivector space. A multivector expressed in that basis then reads, for instance for $d = 3$, $x = x_\emptyset + x_1 e_1 + x_2 e_2 + x_3 e_3 + x_{12} e_1 e_2 + x_{13} e_1 e_3 + x_{23} e_2 e_3 + x_{123} e_1 e_2 e_3$.

The geometric product is fully defined by bilinearity, associativity, and the condition that the geometric product of a vector with itself is equal to its norm. The geometric product generally maps between different grades. For instance, the geometric product of two vectors will consist of a scalar, the inner product between the vectors, and a bivector, which is related to the cross-product of $\mathbb{R}^3$. In particular, the conventional basis elements of grade $k > 1$ are constructed as the geometric product of the vector basis elements $e_i$. For instance, $e_{12} = e_1 e_2$ is a basis bivector. From the defining properties of the geometric products it follows that the geometric product between orthogonal basis elements is antisymmetric, $e_i e_j = -e_j e_i$. Thus, for a $d$-dimensional basis space, there are $\binom{d}{k}$ independent basis elements at grade $k$.

**Projective geometric algebra.**  To represent three-dimensional objects including absolute positions, we use a geometric algebra based on a base space with $d = 4$, adding a *homogeneous coordinate* to the 3D space.[4] We use a basis $(e_0, e_1, e_2, e_3)$ with a metric such that $e_0^2 = 0$ and $e_i^2 = 1$ for $i = 1, 2, 3$. The multivector space is thus $2^4 = 16$-dimensional. This algebra is known as the projective geometric algebra $\mathbb{G}_{3,0,1}$.

**Canonical embedding of geometric primitives.**  In $\mathbb{G}_{3,0,1}$, we can represent geometric primitives as follows:

- Scalars (data that do not transform under translation, rotations, and reflections) are represented as the scalars of the multivectors (grade $k = 0$).
- Oriented planes are represented as vectors ($k = 1$), encoding the plane normal as well as the distance from the origin.
- Lines or directions are represented as bivectors ($k = 2$), encoding the direction as well as the shift from the origin.
- Points or positions are represented as trivectors ($k = 3$).

For more details, we refer the reader to Tbl. 1 in Brehmer et al. [9], or to Dorst [20].

# B  Probabilistic model

Formally, we employ the standard DDPM framework [50] to train a latent variable model $p_\theta(\mathbf{x}_0) = \int p_\theta(\mathbf{x}_{0:T}) d_{\mathbf{x}_{1:T}}$, where $\mathbf{x}_0 = [rsrp, \mathbf{tx}, \mathbf{rx}, \mathbf{mesh}]$ denotes the joint vector of variables following the dataset distribution $p_{data}(\mathbf{x}_0)$. In DDPM, the latent variables $\mathbf{x}_{1:T}$ are noisy versions of the original data, defined by a discrete forward noise process $q(\mathbf{x}_t|\mathbf{x}_{t-1}) = \mathcal{N}\left(\mathbf{x}_t; \sqrt{1 - \beta_t}\mathbf{x}_{t-1}, \beta_t \mathbf{I}\right)$ and $\beta_i > 0$. We approximate the reverse distribution $q(\mathbf{x}_{t-1}\mathbf{x}_t)$ with $p_\theta(\mathbf{x}_{t-1}|\mathbf{x}_t) = \sum_{\hat{\mathbf{x}}_0} q(\mathbf{x}_{t-1}|\mathbf{x}_t, \hat{\mathbf{x}}_0) p_\theta(\hat{\mathbf{x}}_0|\mathbf{x}_t, t)$, where $q(\mathbf{x}_{t-1}|\mathbf{x}_t, \mathbf{x}_0)$ is a normal distribution with closed-form parameters [25]. The forward and backward distributions $q$ and $p$ form a variational auto-encoder [32] which can be trained with a variational lower bound loss. Using the above parametrization of $p_\theta(\mathbf{x}_{t-1}|\mathbf{x}_t)$, however, allows for a simple approximation of this lower bound by training on an MSE objective $\mathcal{L} = \mathbb{E}_{\mathbf{x}_t, \mathbf{x}_0} \left[||f_\theta(\mathbf{x}_t, t) - \mathbf{x}_0||^2\right]$ which resembles denoising score matching [55].

---

[4]A three-dimensional base space is not sufficient to represent absolute positions and translations acting on them in a convenient form. See Brehmer et al. [9], Dorst [20], Ruhe et al. [47] for an in-depth discussion.

To parametrize $p_\theta(\hat{\mathbf{x}}_0|\mathbf{x}_t, t)$, we pass the raw representation of $\mathbf{x}_t$ through the wireless GA tokenizer of Wi-GATr and, additionally, we embed the scalar $t$ through a learned timestep embedding [43]. The embedded timesteps can then be concatenated along the scalar channels in the GA representation in a straightforward manner. Similar to GATr [9], the neural network outputs a prediction in the GA representation, which is subsequently converted to the original latent space. Note that this possibly simplifies the learning problem, as the GA representation is inherently higher dimensional than our diffusion space with the same dimensionality as $\mathbf{x}_0$.

**Equivariant generative modelling.** A diffusion model with an invariant base density and an equivariant denoising network defines an invariant density, but equivariant generative modelling has some subtleties [33]. Because the group of translations is not compact, we cannot define a translation-invariant base density. Previous works have circumvented this issue by performing diffusion in the zero center of gravity subspace of euclidean space [27]. However, we found that directly providing the origin as an additional input to the denoising network also resulted in good performance, at the cost of full E(3) equivariance. We also choose to generate samples in the convention where the $z$-axis represents the direction of gravity and positive $z$ is "up"; we therefore provide this direction of gravity as an additional input to our network.

**Masking strategies.** To improve the performance of conditional sampling, we randomly sample conditioning masks during training which act as an input to the model, as well as a mask on the loss terms. Namely, we sample masks from a discrete distribution with probabilities $p = (0.2, 0.3, 0.2, 0.3)$ corresponding to masks for unconditional, signal, receiver and mesh prediction respectively. If we denote this distribution over masks as $p(m)$, the modified loss function then reads as $\mathcal{L} = \mathbb{E}_{\mathbf{m}\sim p(\mathbf{m}),\mathbf{x}_t,\mathbf{x}_0} \left[||\mathbf{m} \odot f_\theta(\mathbf{x}_t^{\mathbf{m}}, t, \mathbf{m}) - \mathbf{m} \odot \mathbf{x}_0||^2\right]$, where $\mathbf{x}_t^{\mathbf{m}}$ is equal to $\mathbf{x}_0$ along the masked tokens according to $\mathbf{m}$.

# C  Datasets

Table 4 summarizes major characteristics of the two datasets. In the following we explain more details on data splits and generation.

`Wi3R` **dataset.** Based on the layouts of the Wi3Rooms dataset by Orekondy et al. [41], we run simulations for 5000 floor layouts that are split into training (4500), validation (250), and test (250). These validation and test splits thus represent generalization across unseen layouts, transmitter, and receiver locations. From the training set, we keep 10 Rx locations as additional test set to evaluate generalization only across unseen Rx locations. To evaluate the generalization performance, we also introduce an out-of-distribution (OOD) set that features four rooms in each of the 250 floor layouts. In all layouts, the interior walls are made of brick while exterior walls are made of concrete. The The Tx and Rx locations are sampled uniformly within the bounds of the floor layouts (10m $\times$ 5m $\times$ 3m).

`WiPTR` **dataset.** Based on the floor layouts in the ProcTHOR-10k dataset for embodied AI research [18], we extract the 3D mesh information including walls, windows, doors, and door frames. The layouts comprise between 1 to 10 rooms and can cover up to 600 m$^2$. We assign 6 different dielectric materials for different groups of objects (see Tbl. 5). The 3D Tx and Rx locations are randomly sampled within the bounds of the layout. The training data comprises 10k floor layouts, while test and validation sets each contain 1k unseen layouts, Tx, and Rx locations. Again, we introduce an OOD validation set with 5 layouts where we manually remove parts of the walls such that two rooms become connected. While the multi-modality in combination with the ProcTHOR dataset enables further research for joint sensing and communication in wireless, our dataset set is also, to the best of our knowledge, the first large-scale 3D wireless indoor datasets suitable for embodied AI research.

# D  Experiments

## D.1  Predictive modelling

**Models.** We use an Wi-GATr model that is 32 blocks deep and 16 multivector channels in addition to 32 additional scalar channels wide. We use 8 attention heads and multi-query attention. Overall, the model has $1.6 \cdot 10^7$ parameters. These settings were selected by comparing five differently sized networks on an earlier version of the `Wi3R` dataset, though somewhat smaller and bigger networks

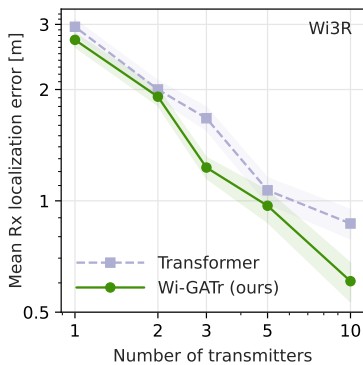

**Figure 6: Rx localization error**, as a function of the number of Tx. Lines and error band show mean and its standard error over 240 measurements.

achieved a similar performance.

Our Transformer model has the same width (translating to 288 channels) and depth as the Wi-GATr model, totalling $16.7 \cdot 10^6$ parameters. These hyperparameters were independently selected by comparing five differently sized networks on an earlier version of the `Wi3R` dataset.

For SEGNN, we use representations of up to $\ell_{max} = 3$, 8 layers, and 128 hidden features. The model has $2.6 \cdot 10^5$ parameters. We selected these parameters in a scan over all three parameters, within the ranges used in Brandstetter et al. [8].

The PLViT model is based on the approach introduced by Hehn et al. [24]. We employ the same centering and rotation strategy as in the original approach around the Tx. Further, we extend the original approach to 3 dimensions by providing the difference in $z$-direction concatenated with the 2D $x$-$y$-distance as one token. Since training from scratch resulted in poor performance, we finetuned a ViT-B-16 model pretrained on ImageNet and keeping only the red channel. This resulted in a model with $85.4 \cdot 10^7$ parameters and also required us to use a fixed image size for each dataset that ensures the entire floor layout is visible in the image data.

**Optimization.** All models are trained on the mean squared error between the model output and the total received power in dBm. We use a batch size of 64 (unless for SEGNN, where we use a smaller batch size due to memory limitations), the Adam optimizer, an initial learning rate of $10^{-3}$, and a cosine annealing scheduler. Models are trained for $5 \cdot 10^5$ steps on the `Wi3R` dataset and for $2 \cdot 10^5$ steps on the `WiPTR` dataset.

**Inference speed.** To quantify the trade-off between inference speed and accuracy of signal prediction, we compare the ray tracing simulation with our machine learning approaches. For this purpose, we evaluate the methods on a single room of the validation set with 2 different Tx locations and two

|  | Wi3R | WiPTR |
|---|---|---|
| Total Channels | 5M | >5.5M |
| Materials | 2 | 6 |
| Transmitters per layout | 5 | 1-15 |
| Receivers per layout | 200 | Up to 200 |
| Floor layouts | 5k | 12k |
| Simulated frequency | 3.5 GHz | 3.5 GHz |
| Reflections | 3 | 6 |
| Transmissions | 1 | 3 |
| Diffractions | 1 | 1 |
| Strongest paths retained | 25 | 25 |
| Antennas | Isotropic | Isotropic |
| Waveform | Sinusoid | Sinusoid |

**Table 4:** Dataset details and simulation settings for dataset generation.

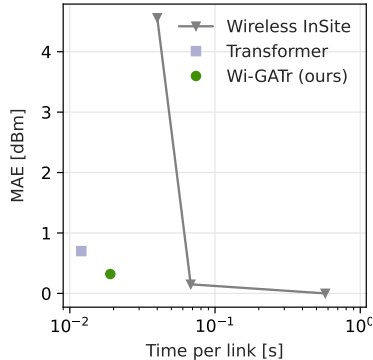

**Figure 7: Inference wall time** vs signal prediction error per Tx/Rx prediction on the first room of the `WiPTR` validation set.

equidistant grids at $z \in \{2.3, 0.3\}$ with each 1637 Rx locations. Figure 7 summarizes the average inference times per link with the corresponding standard deviation. While Wireless InSite $(6/3/1,$ i.e., 6 reflections/3 transmissions/1 diffraction) represents our method that was used to generate the ground truth data, it is also by far the slowest approach. Note that we only measure the inference speed of Wireless InSite for each Tx individually without the preprocessing of the geometry. By reducing the complexity, e.g., reducing the number of allowed reflections or transmissions, of the ray tracing simulation the inference time can be reduced significantly. For example, the configuration $3/2/1$ shows a significant increase in inference speed, but at the same time we can already see that the simulation results do not match the ground truth anymore. This effect is even more pronounced for the case of Wireless InSite $3/1/1$. Our machine learning solutions outperform all tested configurations of Wireless InSite in terms of inference speed, while at the same time keeping competitive performance in terms of prediction accuracy (MAE) compared to the data generation simulation itself in a simpler configuration setting.

In addition, the differentiability of ML approches enables them to solve inverse problems and such as finetuning to real-world measurement data. Finetuning, often referred to as calibration, remains challenging for simulation software and will likely lead to increased MAE as the ground truth is not given by Wireless InSite itself anymore.

## D.2 Probabilistic modelling

**Experiment setup.** For all conditional samples involving $p(F_u|F_k, t, r, h)$, we always choose to set $F_k$ to be the floor and ceiling mesh faces only and $F_u$ to be the remaining geometry. This amounts to completely predicting the exterior walls, as well as the separating walls/doors of the three rooms, whereas the conditioning on $F_k$ acts only as a mean to break equivariance. Since $F$ is always canonicalized in the non-augmented training dataset, this allows for direct comparison of variational lower bounds in Tbl. 3 with the non-equivariant transformer baseline.

**Models.** For both Wi-GATr and the transformer baseline, we follow similar architecture choices as for the predictive models, using an equal amount of attention layers. To make the models timestep-dependent, we additionally employ a standard learnable timestep embedding commonly used in

| Object | Material name |
|---|---|
| Ceiling | ITU Ceiling Board |
| Floor | ITU Floor Board |
| Exterior walls | Concrete |
| Interior walls | ITU Layered Drywall |
| Doors and door frames | ITU Wood |
| Windows | ITU Glass |

**Table 5:** Dielectric material properties of objects in `WiPTR`.

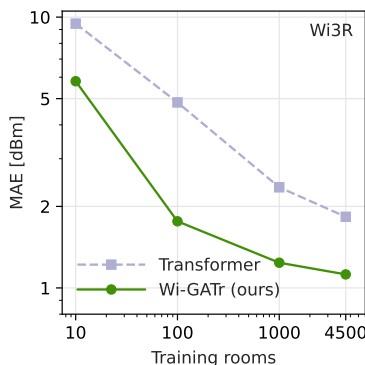

**Figure 8:** Mean absolute errors of received power as a function of number of training rooms for conditional diffusion model samples.

diffusion transformers [43] and concatenate it to the scalar channel dimension.

**Optimization.** We use the Adam optimizer with a learning rate of $10^{-3}$ for the Wi-GATr models. The transformer models required a smaller learning rate for training stability, and thus we chose $3 \cdot 10^{-4}$. In both cases, we linearly anneal the learning rate and train for $7 \cdot 10^5$ steps with a batchsize of $64$ and gradient norm clipping set to $100$.

**Evaluation.** We use the DDIM sampler using 100 timesteps for visualizations in Fig. 5 and for the error analysis in Fig. 8. To evaluate the variational lower bound in Tbl. 3, we follow [39] and evaluate $L_{vlb} := L_0 + L_1 + \dots L_T$, where $L_0 := -\log p_\theta(\mathbf{x}_0|\mathbf{x}_1)$, $L_{t-1} := D_{KL}(q(\mathbf{x}_{t-1}|\mathbf{x}_t,\mathbf{x}_0)||p_\theta(\mathbf{x}_{t-1}|\mathbf{x}_t))$ and $L_T := D_{KL}(q(\mathbf{x}_T|\mathbf{x}_0), p(\mathbf{x}_t))$. To be precise, for each sample $\mathbf{x}_0$ on the test set, we get a single sample $\mathbf{x}_t$ from $q$ and evaluate $L_{vlb}$ accordingly. Table 3 reports the mean of all $L_{vlb}$ evaluations over the test set.

**Additional results.** Fig. 8, shows the quality of samples from $p_\theta(h|F,t,r)$ as a function of the amount of available training data, where we average over 3 samples for each conditioning input. It is worth noting that diffusion samples have a slightly higher error than the predictive models. This shows that the joint probabilistic modelling of the whole scene is a more challenging learning task than a deterministic forward model.

To further evaluate the quality of generated rooms, we analyze how often the model generates walls between the receiver and transmitter, compared to the ground truth. Precisely, we plot the distribution of received power versus the distance of transmitter and receiver in Fig. 9 and color each point according to a line of sight test. We can see that, overall, Wi-GATr has an intersection error of $0.26$, meaning that in $26\%$ of the generated geometries, line of sight was occluded, while the true geometry did not block line of sight between receiver and transmitter. This confirms that the diffusion model correctly correlates the received power and receiver/transmitter positions with physically plausible geometries. While an error of $26\%$ is non-negligible, we note that this task involves generating the whole geometry given only a single measurement of received power, making the problem heavily underspecified. Techniques such as compositional sampling [21] could overcome this limitation by allowing to condition on multiple receiver and received power measurements.

## E   Discussion

Progress in wireless channel modelling is likely to lead to societal impact. Not all of it is positive. The ability to reconstruct details about the propagation environment may have privacy implications. Wireless networks are ubiquitous and could quite literally allow to see through walls. At the same time, we believe that progress in the development of wireless channel models may help to reduce radiation exposure and power consumption of wireless communication systems, and generally contribute to better and more accessible means of communication.

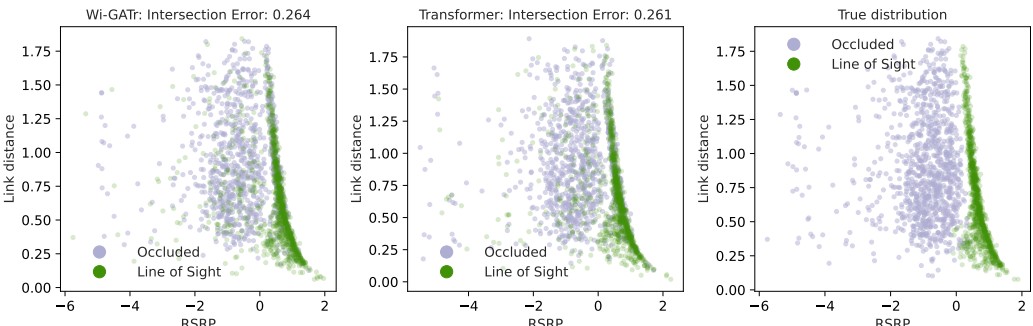

**Figure 9:** A scatter plot of normalized received power versus normalized distance between receiver and transmitter. Each point is colored depending on having line of sight between the receiver and transmitter given the room geometry. Left: The geometry used for calculating line of sight is given by conditional diffusion samples using Wi-GATr. Middle: The geometry used for calculating line of sight is given by transformer samples. Right: The geometry used for calculating line of sight is taken from the test data distribution.

