# OpenReview forum: "Probabilistic and Differentiable Wireless Simulation with Geometric Transformers"
_NeurIPS.cc/2024/Conference — Submitted to NeurIPS 2024_

### Official Review · Reviewer_yJCq · 2024-07-01

**Soundness:** 2
**Presentation:** 3
**Contribution:** 2
**Rating:** 5
**Confidence:** 3

**Summary:**

The paper proposes the use of the Wireless Geometric Algebra Transformer (Wi-GATr) to model signal propagation. Based on the Wi-GATr network, it introduces a differentiable prediction model and a diffusion model. Compared to traditional statistical and ray-tracing methods, the proposed approach not only addresses conventional signal prediction problems but also tackles inverse problems such as receiver localization and 3D environment reconstruction. Experimental results demonstrate the effectiveness of this method. Furthermore, the authors present two large-scale wireless signal propagation datasets.

**Strengths:**

1) The perspective is interesting. This paper models wireless propagation as a probability model based on diffusion, thus offering a unified approach to address signal prediction and inverse problems such as receiver localization or 3D scene reconstruction.
2) The paper is logically structured, with a comprehensive background introduction and high readability.

**Weaknesses:**

1) There is a gap between the challenges and the solutions. The author asserts that wireless surrogate modelling faces challenges like data scarcity and diverse data types. However, the lack of analysis on these issues makes the proposed solutions appear abrupt. It is recommended to provide insights that lead to the proposed solutions of this paper.
2) The innovation is somewhat limited. Wi-GATr primarily extends the GATr method into a wireless setting, with equivariance being a pre-existing property of the original framework. Apart from tokenizing input data, did the paper introduce any additional advancements? It would be beneficial for the authors to highlight these aspects.
3) The experimental evaluation is not sufficiently convincing. For more details, please refer to the "Questions*" part.
4) There are some typographical errors in the paper. For example, lines 56 and 57 do not correspond to Figure 1. Additionally, the abstract mentions transmitter localization, but the main text describes receiver localization, among other discrepancies.

**Questions:**

1) The contributions of Wi-GATr appear to be limited. What are the specific contributions of Wi-GATr, and how do they compare in detail with those of GATr?
2) In the background section, the authors mention a series of neural statistical models [19, 40, 42, 56] and neural ray tracers [29, 41, 58], claiming the proposed method is "sample-efficient and generalizes to novel scenes". However, the evaluation section only compares two methods, which seems relatively weak. What considerations are made when choosing the comparison methods?
3) In qualitative experiments, the proposed method is compared with ViT. Why is it not compared with the benchmarks used in quantitative experiments—SEGNN and PLViT?
4) The experimental results in Figure 5 lack the necessary explanations. For instance, does the "Geometry reconstruction" on the right side lack comparison with ground truth?
5) This paper introduces both differentiable prediction models and diffusion models, claiming that diffusion models overcome the limitations of differentiable prediction models. Could the author conduct experiments to compare these two models and validate this claim?

**Limitations:**

Sufficiently discussed.

---

> ### Author Rebuttal · Authors · 2024-08-07
>
> We thank reviewer yJCq for providing a detailed review of our paper.
> We are happy to read that they found it logically structured and that they appreciated the novel treatment of forward and inverse problems with a diffusion model.
> Now, we address reviewer yJCq’s concerns and also indicate if it is shared by fellow reviewers.
>
> **Limited innovation... contributions of our WiGATr vs. GATr [9] .... any additional advancements?**
> We believe it is unfair making a direct contribution comparison of our work with GATr [9], since both address different problems.
> GATr [9] tackles the problem of representing geometric data and introduces a novel architecture.
> In contrast, our work tackles a differentiable wireless simulation problem of prediciting wireless signal characteristics from 3D scenes.
> 3D scenes are inherently geometric, and to the best of our knowledge, we are the first to exploit relevant symmetries and find GATr well-suited for this task.
> Apart from our model contributions (e.g., tokenization, generative formulation), we tackle inverse problems in wireless (e.g., receiver localization) and propose two new large-scale datasets.
>
> **Paper mentions neural statistical models [19, 40, 42, 56] and neural ray tracers [29, 41, 58] ... however evaluation section only compares two methods ... considerations made when choosing baselines?**
> We choose baselines that tackles the specific problem considered in the paper: jointly modelling the 2d/3d representations (e.g., mesh, image) of the propagation environment and wireless characteristics (e.g., path loss, receive signal strength).
> Neural statistical models [19, 40, 42, 56] are not applicable since they typically ignore scene representations.
> Neural ray tracers [29, 41, 58] are more related to our work.
> We now introduce Sionna RT [29] as a baseline (see Fig. 3 in rebuttal pdf; faster than ours, but larger errors).
> The other two are not applicable: [41] does not generalize to novel scenes and [58] does not consider geometric representation.
> As a result, we now compare against _five_ relevant baselines that jointly model the 2d/3d representation of the scene and wireless characteristics: PLViT, SEGNN, Transformer, Sionna-RT*, Naive Transformer* (* = based on suggestions from reviewers).
> In all cases, we find our previous results hold: our approach outperforms all baselines in accuracy of predicted signal strengths.
>
> **Gap between the challenges and the solutions ... lack of analysis on these issues makes the proposed solutions appear abrupt ... recommend to provide insights that lead to the proposed solution**
> Thanks for the suggestion. We will use the extra page in final version to connect the challenges to our proposed solutions, e.g., the inductive biases improve data-efficiency, transformer tokenization scheme facilitates processing of 3D geometry.
>
> **Qualitative comparison with ViT ... quantitative with PLViT and SEGNN ... why the difference**
> Good catch, the "ViT" label in Fig. 1 is a typo and should read "PLViT". We will fix this for the next version.
> We were able to evaluate SEGNN only on simpler datasets (only Wi3R) due to its memory requirements.
>
> **Fig. 5 lacks necessary explanations**
> Due to space constraints, we moved some experimental details for our probabilistic model to the Appendix D.2. Furthermore, Figure 9 compares the behavior of generated geometries with the ground truth. In Figure 5 (c), we omit a direct comparison to the ground truth, as this inverse problem admits a variety of solutions. Instead, we visualize two diffusion samples conditioned on two different values for the signal strength respectively. As the signal strength decreases, we observe that the model is more likely to occlude line of sight between the receiver and transmitter locations, which mimics the dynamics of the training dataset.
>
> **This paper introduces both differentiable prediction models and diffusion models, claiming that diffusion models overcome the limitations of differentiable prediction models. Could the author conduct experiments to compare these two models and validate this claim?**
> Due to their different learning objectives, diffusion and prediction models are not easily comparable directly in terms of mean absolute errors. For example, when solving the inverse problem of receiver localization, the diffusion model learns a distribution of points (see Fig. 5b), whereas the SGD-based solution with the prediction model can only recover a single mode of this distribution. This inherent measure of uncertainty is one of the main strengths of diffusion over predictive modelling. However, when learning unimodal distributions (such as signal prediction, which is assumed to be deterministic in our setup), we have confirmed that the diffusion model performs comparably to the predictive model, as can be seen when comparing Fig. 8 with Fig. 3.
>
> **Typographical errors ... L56 L57 ... transmitter localization**
> Thanks for pointing them out. We will fix them.
>
> We thank the reviewer again for their detailed feedback. We hope we were able to address their concerns and look forward to discussing further.

---

> > ### Comment · Reviewer_yJCq · 2024-08-12
> >
> > This response clearly distinguishes this work from GATr and highlights its innovative aspects. Additionally, the authors have refined the previously unclear expression and added Sionna-RT as a new competitor. However, this comparison is included only in the qualitative and not the quantitative experiments. It is also suggested that comparisons with the naive transformer be included in all experiments. Based on the above, I am inclined to raise my score to Borderline Accept.

---

> > > ### Author Response · Authors · 2024-08-12
> > >
> > > We thank the reviewer for the response and are happy to hear they appreciated our new comparisons.
> > >
> > >
> > > We would like to correct one detail: we also ran quantitative experiments with Sionna-RT. These results are included in the rebuttal text and in Fig. R2 of the rebuttal pdf. Based on those results, we conclude that Sionna-RT (in its current state) is not capable of solving the tasks discussed in the paper due to lack of transmission models.

---

### Official Review · Reviewer_6Syy · 2024-07-09

**Soundness:** 3
**Presentation:** 3
**Contribution:** 3
**Rating:** 4
**Confidence:** 4

**Summary:**

This paper proposes the use of a transformer architecture to model electromagnetic propagation of physical systems. The approach is claimed to outperform existing methods by (i) computational efficiency (compared to raytracers) and (ii) enabling solving inverse problems. The method is evaluated on a number of benchmark tasks and the paper is accompanied by two new datasets.

**Strengths:**

The idea of modeling electromagnetic wave propagation using transformer architectures is novel and interesting.

The generality of the approach enables a large number of tasks in wireless communication systems that would otherwise the use of raytracers or other electromagnetic modeling software.

Large parts of the main body of the paper are well-written and easy to follow.

**Weaknesses:**

The main body of the paper lacks the details of the proposed pipeline and transformer architecture. In fact, all of the interesting and technical details are relegated to the appendices.

One of the main motivations of the paper seems to be that most raytracers are too slow. However, the authors seem to ignore recent projects, such as Instant RM (https://github.com/NVlabs/instant-rm) which can compute coverage maps in a few milliseconds, depending on the desired accuracy.

It is unclear why [29] is cited as a non-differentiable raytracer although it is, to my knowledge, the only raytracer that actually is. Instant RM is also differentiable and calibration results for both tools were already demonstrated. To be honest, I have the impression that the authors tried to cover up the fact that [29] is a powerful *differentiable* raytracer that enables solving inverse problems.

Although the authors claim that channel impulse responses can be generated, this is not demonstrated in the paper. I think that this claim should be removed unless the authors demonstrate that it is actually feasible.

The description of scene geometry recovery is incomprehensible to me.

In Figure 7, the Wi-GATr is around 20ms for inference for a tiny indoor scene. The authors should compare this against Instant RM which can probably run even faster and is differentiable.

It would be good to get confidence information (e.g., standard deviation) in Fig. 3 and Fig. 4.

**Questions:**

The authors should comment on the scalability of the proposed method to scenes with millions of triangles.

How well would a neural-network-based positioning pipeline do on the task used to generate Figure 4?

**Limitations:**

The paper lacks a detailed comparison to the capabilities of the differential raytracer from [29]. In fact, I feel that for the individual tasks, more baselines should be included.

Scalability to very large datasets and extremely complex scenes is unclear.

It is unclear whether the method generalizes to electromagnetic environments that are nonreciprocal (e.g., containing certain nonreciprocal metasurfaces).

It is unclear whether the method generalizes to scenarios in which ray-tracing is inaccurate (e.g., scenarios at low carrier frequencies).

---

> ### Author Rebuttal · Authors · 2024-08-07
>
> We thank reviewer 6syy for providing a detailed review of our paper.
> We are glad that they appreciated the generality of the approach, and in particular that they found our paper well-written and easy to follow.
> Now, we address reviewer 6syy’s concerns and also indicate if it is shared by fellow reviewers.
>
> **Several fast and differentiable ray tracers ... such as Sionna RT exist ... comparison missing*** (common concern - 6Syy, yJCq)
> Thanks for the suggestion. For this rebuttal, we provide additional experiments about Sionna RT (see Figs. R2 and R3 of the rebuttal pdf).
> We will add this result to our appendix.
> Although Sionna RT is faster than our approach, we observe large errors (12 dB without calibration, 8 dB with calibration) on the WiPTR validation set, compared to the proposed approach (0.74 dB).
> This is mainly because Sionna does not implement a model for transmission/refraction. Its predictions for the received power behind walls are therefore systematically too low. It also leads to faster computation as rays split at refraction events and the complexity of the ray tracing increases.
> In constrast, our approach is fully data-driven, not limited to certain physics models, and is thus able to learn interaction phenomena, including transmission, from the training data.
>
> **Unclear why Sionna-RT [29] is cited as a non-differentiable raytracer**
> Thank you for the catch, this (line 26) is a typo. We apologize and will fix it. All other references to Sionna-RT (e.g., line 92) mention differentiability.
>
> **Main motivations of the paper ... most raytracers are too slow ... ignores Instant RM which computes coverage maps in a few milliseconds**
> InstantRM is concurrent work -- the code was released 20 days before submission deadline, and publicized on a blog [A] 1 month after submission deadline.
> We are glad to cite, discuss and potentially compare in the next version of the paper.
> Yet, the models used in InstantRM are more limited than Sionna RT in terms of accuracy. Our arguments about Sionna RT (lack of transmission) apply thus to InstantRM as well.
>
> [A] https://developer.nvidia.com/blog/fast-and-differentiable-radio-maps-with-nvidia-instant-rm/
>
> **Authors claim that channel impulse responses can be generated, this is not demonstrated in the paper**
> We appreciate that the reviewer points this out. We explicitly mention modeling of received power (e.g., lines 12, 50, 56).
> We would like to reduce potential ambiguity in this regard. Could the reviewer please point us to the exact statement that seems like we are claiming that we are generating the full channel impulse response?
>
> **Description of scene geometry recovery is incomprehensible**
> Thanks for pointing this out. We will rewrite this section and improve clarity.
>
> **Add confidence information (e.g., standard deviation) in Fig. 3 and Fig. 4**
> Thanks for the suggestion. We agree and will add measures of uncertainty to our results for the final version.
>
> **Scalability to very large datasets and extremely complex scenes is unclear** (common concern - LEve, 6Syy)
> We already evaluate our approach on a large test dataset (e.g., 1K diverse scenes in WiPTR proposed in this paper, $>5.5$M channels in total) -- this is significantly larger than previous efforts.
> In addition, we ran scalability experiments and found proming results.
> As shown in Fig. R4 in the rebuttal pdf, WiGATr remains at sub-second latency until $\sim$10k mesh faces.
> Although scenes with 500k mesh faces take up to an hour with our vanilla implementation, this is faster than a conventional, GPU-optimized ray tracer based on preliminary experiments.
>
> **Unclear whether the method generalizes to electromagnetic environments that are nonreciprocal**
> This is a great observation. By default, the equivariance properties of our approaches consider reciprocity.
> However, if necessary, this can be locally disabled by providing additional orientation information that breaks the symmetry of the problem.
>
> **Unclear whether the method generalizes to scenarios in which ray-tracing is inaccurate**
> The accuracy of our approach depends on the accuracy of the underlying training data.
> As long as the training data is generated by ray tracing, data-driven models cannot compensate for this shortcoming.
> However, Wi-GATr could also be trained or finetuned on measurements and then yield accurate predictions in scenarios where ray tracing is inaccurate.
>
> **How well would a neural-network-based positioning pipeline do on the task used to generate Figure 4?**
> To the best of our knowledge, existing positioning pipelines cannot be used in our task since they assume measurements from the identical environment at both training and test time.
> In contrast, our approach is "zero-shot": positioning is evaluated on unseen environments at test time.
>
> We thank the reviewer again for their constructive feedback. We hope we were able to address their concerns and look forward to discussing further.

---

> > ### Comment · Reviewer_6Syy · 2024-08-12
> >
> > Thanks a lot for the clarifications; I have two remaining comments:
> >
> > - Scalability remains to be a real issue. The statement that the proposed scheme is faster than ray tracers for large scenes seems incorrect. The computation of coverage maps is typically done using shoot-and-bounce, whose complexity is almost independent of the number of primitives in a scene.
> >
> > - The fact that Sionna RT, Instant RM, or any other ray tracer does not implement refraction does not justify the proposed approach. In fact, adding refraction could be added without any problem. Thus, I do not see a strong motivation apart from speed (which is a bit thin since fast raytracers do exist).

---

> > > ### Author Response · Authors · 2024-08-13
> > >
> > > **... The computation of coverage maps is typically done using shoot-and-bounce, whose complexity is almost independent of the number of primitives in a scene.**
> > >
> > > **This is incorrect.** The complexity of shoot-and-bounce (SBR) algorithms is *not* independent of the number of primitives in a scene if you include refraction and diffraction. In fact, it highly depends on the geometry of the scene.
> > > * Refraction: At every refraction event, the incoming ray is split into two outcoming rays, thus increasing the complexity. One that is reflected and one that is refracted.
> > > * Diffraction: Diffraction first requires finding edges in the scene. The number of edges increases with the number faces in the scene unless neighboring faces are entirely parallel (which they are not in practice).
> > >
> > > As a result, it is common practice to limit the number of diffractions and refractions during ray tracing to find a trade-off between accuracy and scalability. For example, Sionna resorts to only modeling a single diffraction event at the end of an interaction chain. InstantRM (a fast SBR ray tracer) does not support diffraction. As mentioned before, none of them model refractions.
> > >
> > > In contrast, the run time and scalability of our method is independent of the exact geometry of the mesh.
> > >
> > > &nbsp;
> > >
> > > **... In fact, adding refraction *could be added* [to ray tracers] without any problem.**
> > >
> > > **Our method should be compared to the actual methods available today**, even if ray tracers can in theory be extended to include refraction.
> > > As mentioned above, even if they only predict coverage maps, the extension of ray tracers is not trivial. From personal interaction with Sionna RT authors, we have learned that implementing differentiable transmission efficiently is challenging.
> > >
> > >
> > > Even more so: To the best of our knowledge, there are no works that ever showed the feasibility of optimization of the transmitter or receiver position using a ray tracer, neither on real or on simulated data. Sionna RT have claimed that it is possible, but we have not found an experiment to show it (only transmitter orientation). We show we can solve receiver localization.

---

### Official Review · Reviewer_2AZ5 · 2024-07-12

**Soundness:** 2
**Presentation:** 2
**Contribution:** 2
**Rating:** 3
**Confidence:** 4

**Summary:**

Authors proposed transformer based ideas on very well studied area: Wireless environment simulation.
The key idea here is to capitalize Geometric Transformers to simulate radio environments. It true that wireless (directional) signal propagation is a ray tracing approach, meaning a highly directional wireless signal (Ray) may bounce off ambient surfaces or directly reach a receiver. The proposed method inserts geometric shapes in the environment as tokens in a transformer networks. The trained transformer predicts the received power at a given point in 3D space. Transformer is trained and evaluated using two datasets: Wi3R and WiPTR that simulate indoor signal propagation environments. In comparison to the baselines, transformer architecture requires 20 times less data.

**Strengths:**

1. It is interesting idea to see if transformer architectures are suitable to model radio propagation environments
2.  Propagation models consider the material of ambient surface, antenna parameters, location of transmitter and receiver

**Weaknesses:**

1. It is not clear if datasets are of any relevance to real world environment, since the primary challenge of any modelling problem is simulation to reality gap. Since wireless channel modelling is very well studied area, a novel contribution must take in to account such differences rather than results that show the ability of transformer architecture in modelling a wireless environment
2. There are several high fidelity radio propagation modelling software, perhaps it is important to consider datasets generated from such model, current evaluation is very limited and primitive. Fig 2 and 5 are no comparison to robust channel models that are available to wireless researcher and practitioners.
3. The modelling of radio environment is no clear, reviewer is of impression that several affects like diffraction, refraction are not considered in the datasets
4. The paper also has weakness: WiNeRT: Towards Neural Ray Tracing for Wireless Channel Modelling and Differentiable Simulations which is both papers have not considered user mobility: coherence time, coherence bandwidth
5. Current evaluation is only limited to indoor environments
6. Since the prior work has already established neural network architecture are useful modelling, to push the state of the art, it is important to show accurate modelling than yet another architecture to model wireless channel.
7. Upon inspecting table 2, the reviewer is afraid that there might be issue with results here. There is 80dBm difference is accuracy with transformers based modelling, usually such a difference is unacceptable, can author please explain the training of transformer model and why it produced such a large error. The reviewer is concerned that whether such sample point is a fair to benchmark againt

**Questions:**

1. The challenging problem of wireless environment simulation is how the simulation and real world measurement hold together.
2. Since, this is very old problem, the novel solution must show the ability to bridge gap between simulation and real channel traces
3. It is quite important show how well the proposed method simulates various wireless environment, currently such evaluation is missing.
4. Could you please provide any benchmarks with real-world channel traces datasets?

**Limitations:**

In reviewer's opinion authors have not sufficiently addressed all the limitations of the current work. I encourage authors to look at weakness section and update the limitations of the work in the current draft

---

> ### Author Rebuttal · Authors · 2024-08-07
>
> We thank reviewer 2AZ5 for providing a detailed review of our paper.
> Overall, we are glad that the reviewer finds using transformers for radio propagation modeling an interesting idea and appreciates that we include relevant information in our environment input.
> Now, we address reviewer 2AZ5’s concerns and also indicate if it is a common concern shared by fellow reviewers.
>
> **Problem is simulation to reality gap ... relevance to real-world environment**
> Indeed, mitigating the simulation-to-reality gap is an important problem. However, we find it complementary to the problem considered in this paper: modelling wireless characteristics of novel 3D propagation environments by exploiting symmetries. We believe solving this problem in a large-scale controlled setting (be it simulation or real) is a precursor to the problem of simulation-to-reality gap.
>
> **Benchmarks with real-world data ... is missing**  (common concern - reviewers LEVe, 2AZ5)
> Firstly, we remark that evaluations on simulated data have their own merits: it allows evaluation on a large-scale (e.g., diverse environments, locations of transmitter/receiver) in a controlled setting and enables better analysis of approaches and results.
> While we agree with reviewers that it would be ideal to extend our evaluation to real-world datasets, we are limited by their availability.
> To the best of our knowledge, no relevant large-scale real-world dataset exist for the problem we tackle and hence previous works [29, 41, 35, 25] have also largely relied on simulated datasets.
> In fact, even the simulated datasets that exist (e.g., Wi3Rooms [41], Etoile/Munich [29]) are small-scale with low diversity in terms of 3D environments.
> This lack of high-quality wireless datasets with diverse scenes motivated us to generate and release two new datasets with this paper.
> Our symmetry considerations apply in the real-world and in simulation, thus our key message - that wireless channel modelling is a geometric problem and one should use symmetry-aware architectures and algorithms to solve it - should hold both in simulation and real-world measurements.
>
> **How well proposed method simulates various wireless environment ... evaluation is missing**
> We are unclear on what "various" corresponds to here and would appreciate the reviewer's clairification. Nonetheless, we believe we evaluate our approach in a fairly diverse setup (e.g., >1K novel scenes, i.i.d.\ 3D locations of tx/rx) improving upon prior work (e.g., [41] on 1 scene, [29] on 1 scene, [58] evaluates on 50 scenes). This is inline with comments of other reviewers e.g., "thorough evaluation ... various tasks ... multiple baselines" (LEve)
>
> **Several high fidelity radio propagation modelling software ... important to consider datasets from such models**
> We use the commercial ray-tracer "Remcom Wireless InSite" (see Sec. 4 in the paper) that is popular in the literature ([41, 3]).
> Please let us know your concrete concerns or missing features of this software.
>
> **Modelling of radio environment is not clear ... diffration/refraction not considered**
> This is incorrect. All simulated paths include diffraction, transmission, and reflection effects; see Table 4 in Appendix C for details. We will clarify in the main paper.
>
> **Mobility is not considered**
> Mobility is an interesting and relevant wireless channel modeling research problem. However, this complements our work that studies on generalization and symmetries of novel 3D scenes.
>
> **Evaluation limited to indoor environments** (common concern - reviewers LEve, 2AZ5)
> Evaluation on outdoor scenarios would make a great addition.
> However, the physical phenomenons (e.g., diffraction, reflections) of wireless signals would largely remain the same as a function of the 3D geometry in both indoor or outdoor settings.
> Nonetheless, in this paper, we choose indoor scenarios since we believe it is a more challenging scenario to evaluate influence of 3D scenes towards wireless characteristics.
> For instance, heights of surfaces play a bigger role (since signals can reflect off floors and ceilings) and transmissions through surfaces have significant contributions towards receive powers in non-LOS regions.
>
> **NNs already shown to be useful for modelling wireless propagation ... how accurate is this yet another architecture**
> Indeed, ML-based approaches are shown to be successful for modelling wireless propagation. These studies are often limited to 2D representations of the scene (e.g., binarized satellite images). However, wireless propagation is inherently influenced by the 3D structure of the propagation environment and prior approaches exhibit large errors in these scenarios (details in Sec. 5.1). In contrast, our novel equivariant architecture exploits certain symmetries of the 3D environment and significantly outperforms prior relevant ML-based approaches.
>
> **Table 2 ... 80 dBm error ... explain large error of baseline**
> Thank you for highlighting this result. It is one of the main arguments of our paper. The baseline is strong on *in-domain* data (e.g., MAE of 1.32 dB on the unseen floor plans, see Table 2).
> However, the generalization to *out-of-distribution* data is a significant failure mode (e.g., MAE of 78.68 dB on rotated data).
> In contrast, our proposed Wi-GATr architecture improves the robustness under domain shifts by incorporating the symmetries of the problem. By construction, it is robust to translations and rotations of the scene; we also find an improved robustness to other out-of-distribution test sets.
>
> We thank the reviewer again for the thorough review. We hope we were able to clear up some concerns and look forward to discussing more.

---

> > ### Comment · Reviewer_2AZ5 · 2024-08-12
> >
> > Thank you very much for your response.
> >
> > **Benchmarks with real-world data ... is missing (common concern - reviewers LEVe, 2AZ5)**
> > Authors may find the following resources valuable for further evaluation
> >
> > https://nvlabs.github.io/sionna/ : Very large scale simulated dataset generation feasible
> > https://www.deepsig.ai/datasets/ :  Simulated/Emulated dataset for use
> > https://www.deepsense6g.net/: Large scale real world datasets available
> >
> >
> > **We use the commercial ray-tracer "Remcom Wireless InSite" (see Sec. 4 in the paper) that is popular in the literature ([41, 3]). Please let us know your concrete concerns or missing features of this software.**
> >
> > In the interest of community as a whole, it might be a good idea to use accessible and open-simulators, since we may not access to them. Could you please present a comparative analysis Sionna simulator vs Remcom.  Also, how closely, a particular scenario has been simulated, scatters, reflections, material properties, mobility scenarios(I believe they are currently missing)
> >
> >
> > Modelling of radio environment is not clear ... diffration/refraction not considered
> > This is incorrect. All simulated paths include diffraction, transmission, and reflection effects; see Table 4 in Appendix C for details. We will clarify in the main paper.
> >
> > **Mobility is not considered
> > Mobility is an interesting and relevant wireless channel modeling research problem. However, this complements our work that studies on generalization and symmetries of novel 3D scenes.**
> > Could you please elaborate, I couldn't understand the above
> >
> >
> > **Evaluation limited to indoor environments (common concern - reviewers LEve, 2AZ5)**
> > Evaluation on outdoor scenarios would make a great addition. However, the physical phenomenons (e.g., diffraction, reflections) of wireless signals would largely remain the same as a function of the 3D geometry in both indoor or outdoor settings. Nonetheless, in this paper, we choose indoor scenarios since we believe it is a more challenging scenario to evaluate influence of 3D scenes towards wireless characteristics. For instance, heights of surfaces play a bigger role (since signals can reflect off floors and ceilings) and transmissions through surfaces have significant contributions towards receive powers in non-LOS regions.
> >
> > *Since, reviewer thinks other way, could you please provide any references in the draft supporting the above*
> >
> >
> > **Table 2 ... 80 dBm error ... explain large error of baseline**
> > Thank you for highlighting this result. It is one of the main arguments of our paper. The baseline is strong on in-domain data (e.g., MAE of 1.32 dB on the unseen floor plans, see Table 2). However, the generalization to out-of-distribution data is a significant failure mode (e.g., MAE of 78.68 dB on rotated data). In contrast, our proposed Wi-GATr architecture improves the robustness under domain shifts by incorporating the symmetries of the problem. By construction, it is robust to translations and rotations of the scene; we also find an improved robustness to other out-of-distribution test sets.
> > Could you please provide a little more information, on how many experiments are performed and what is OOD data looks like.

---

> > > ### Author Response · Authors · 2024-08-13
> > >
> > > **[Real-world datasets] Authors may find the following resources valuable for further evaluation**
> > >
> > > Thanks for pointing out the resources. However, they do not apply since they are either simulated, small-scale, or designed for a different task. This further strengthens our argument that real-world data is lacking.
> > >
> > > * https://nvlabs.github.io/sionna/ Sionna is a simulator not a real-world dataset. It has been sufficiently discussed in our rebuttal.
> > > * https://www.deepsig.ai/datasets/ As you mention, these are also simulated datasets. On this website, the authors advise against using this data unless for historical or educational purposes.
> > > * https://www.deepsense6g.net/: While being real-data, DeepSense6G tackles a different problem, i.e., beam and blockage prediction at mmWave. For this purpose, much simpler models are sufficient (see https://arxiv.org/abs/2401.17781, published at ICC 2024). Further, only a subset of the scenarios contain 3D information from LiDAR, and none of them contain meshes and material information. It is a different problem and not suitable to show the benefits of equivariant architectures.
> > >
> > > &nbsp;
> > >
> > > **Could you please present a comparative analysis Sionna simulator vs Remcom.**
> > >
> > > We have provided a detailed comparison to Sionna throughout this rebuttal. The summary is that Sionna currently provides simplistic models for physical effects, e.g., lack transmission, single difraction only, no thickness of materials modeled. See Figs. R2 and R3 in the rebuttal pdf for quantitative and qualitative comparisons.
> > >
> > > &nbsp;
> > >
> > > **In the interest of community as a whole, it might be a good idea to use accessible and open-simulators, since we may not access to them**
> > >
> > > With our plans to publish our dataset, we want to make our high fidelity simulations freely available to the community.
> > >
> > > &nbsp;
> > >
> > > **[Mobility is not considered] Could you please elaborate, I couldn't understand the above**
> > >
> > > Mobility is an important problem, but it is not the only problem in wireless channel modeling. We tackle a different problem.
> > >
> > > &nbsp;
> > >
> > > **[Indoor vs outdoor scenarios] Since, reviewer thinks other way, could you please provide any references in the draft supporting the above**
> > >
> > > This is a quote from a very recently accepted challenge at ICASSP 2025 (https://indoorradiomapchallenge.github.io/index.html)
> > > > it's necessary to develop models tailored for indoor environments. In such cases, the refracted electromagnetic field components through obstacles play a more significant role in radio signal propagation, as opposed to the outdoor scenarios that is dominated by reflected field components. Therefore, accurate indoor radio map estimation requires accounting for the larger variety of construction materials and their electromagnetic properties.
> > >
> > > &nbsp;
> > >
> > > **Could you please provide a little more information, on how many experiments are performed and what is OOD data looks like.**
> > >
> > > We have provided details about our OOD data in our paper (Sec. 4) and its appendix (Appendix C).

---

### Official Review · Reviewer_3vGS · 2024-07-14

**Soundness:** 3
**Presentation:** 2
**Contribution:** 2
**Rating:** 5
**Confidence:** 4

**Summary:**

The motivation for this work is that modeling the propagation of electromagnetic signals is critical for designing modern communication systems. Ray tracing simulators are not suitable for inverse problems or integration as channel models in designing communication systems.

In this context, the goal of this paper is to model the interplay between the 3D environment F, transmitting and receiving antennas (each characterized by a 3D position, orientation, and specific antenna characteristics) represented by t and r, respectively, and the signal h between each transmitter and receiver. The 3D geometry F is represented by a triangular mesh, where each triangle is assigned a material type from predefined classes, modeling both the shape and materials of the environment. Once the model is learned, three tasks can be performed:
1. Prediction of the received signal p(h∣F,t,r): The model is trained for this task. At test time, the network can predict signals in unseen, novel scenes. This approach is faster, fine-tunable on real measurements. The model obtained is also differentiable. This is referred to as the forward problem.
1. Localization of the receiver p(r∣F,t,h).
1. Sensing the environment p(F∣t,r,h).

The last two tasks are referred to as inverse problems. The model introduced is an adaptation of the Geometric Algebra Transformer called Wireless (Wi-GATr), used for simulating wireless propagation in a 3D environment. The authors also cast this problem as a generative modeling task of the joint distribution p(F,t,r,h) (from which the above three tasks can be accomplished) using Denoising Diffusion Probabilistic Models and Wi-GATr.

**Strengths:**

The main contributions of this work are as follows:
1. Introducing a new tokenization method for geometric wireless communication environments and transmitter and receiver characteristics.
1. Integrating diffusion-based models with Wi-GATr to model the wireless environment as a generative model, thereby determining the joint distribution of F, t, r, and h.
1. Providing new, larger datasets for the wireless environment modeling to the research community.

**Weaknesses:**

As per my understanding, the main weaknesses of the work are:
1. The novelty of the work lies in tokenizing various geometric objects encountered in the wireless communication scene. However, the same tokenization is used in the vanilla transformer, making it unclear if the new tokenization provides any benefit.
1. As the authors point out, the channel is modeled only in terms of time-averaged non-coherent received power, missing crucial information such as time and direction of arrival, which are essential for modeling wireless environments.
1. While the proposed solutions seem general, most results are presented for the single antenna case. Additionally, the dataset includes only transmitting sinusoidal waveforms, which is limiting as it does not cover larger bandwidths. The wave propagation depends on frequency, and non-linearities can occur with wider bandwidths.

**Questions:**

In addition to the weaknesses listed above, I have the following questions:

1. The authors should clarify how many transmit antennas were used in receiver localization and sensing, and why they believe the results are promising despite the limitations of using a larger number of antennas and ignoring time and angle of arrival information.
1. The authors should also clarify the challenges involved in their novel tokenization and in developing diffusion-based models with the Wi-GATr model.

**Limitations:**

The authors mention limitations in the Discussion section (Section 6).

---

> ### Author Rebuttal · Authors · 2024-08-07
>
> We thank reviewer 3vGS for providing a detailed review of our paper.
> Overall, we are glad that the reviewer appreciates our novel wireless tokenization scheme, our diffusion approach, and that we contribute a novel large-scale dataset to the machine learning community.
> Now, we address reviewer 3vGS’s concerns.
>
> **Clarify the challenges ... novel tokenization**
> The main challenge is the representation of diverse geometric types involved in wireless scenes: points, rays, orientations, and surfaces.
> Most popular representations of 3D geometry are designed for either point clouds or meshes or rays, but not their combination.
> In constrast, our tokenization scheme provides a unified way of representing different combinations of diverse geometric types.
>
> **Same tokenization used both in WiGATr and vanilla transformer baseline ... benefits of tokenization unclear**
> Great suggestion, we indeed did not clearly demonstrate the benefits of the new tokenization scheme in the original paper. We add this ablation now and find promising results. In Fig. R1 in the rebuttal result page, we compare a new "naive transformer" model that does not use our geometric wireless tokenizer to the existing "transformer" result based on the tokenizer. We find that tokenization has a drastic impact on performance: for instance, when training on 1000 rooms, using our wireless tokenizer improves the transformer performance from 2.9 dB to 1.7 dB mean average error. We will provide additional details for this experiment in the final version of the paper.
>
> **Clarify how many transmit antennas were used in receiver localization and sensing**
> We study SISO omni-directional antennas throughout the paper.
> We vary the number of SISO transmitters used for receiver localization in Fig. 4 of the paper.
>
> **Clarify why results are promising despite the limitations of using a larger number of antennas and ignoring time and angle of arrival information**
> We have for now focused on SISO received power prediction as an evaluation task as it has been discussed in the relevant literature before. Adaptation of the tokenization scheme is an interesting idea, as well as extendening the output other channel characteristics. Our considerations about symmetries of the underlying physics still apply.
>
> **Clarify the challenges ... diffusion-based models with the Wi-GATr model**
> One of the main challenges was in achieving a good performance on inverse problem solving using our diffusion model. To maximize the performance of both joint (unconditional) inference as well as marginal (conditional) predictions within the same model, one needs to tune the trade-off between unconditional and conditional log likelihood loss terms during training. Finally, the use of the DDIM scheduler proved to be crucial for achieving sufficiently fast sample generation during evaluation.
>
> **only in terms of time-averaged non-coherent received power ... missing time and direction of arrival**
> In our experiments, we indeed restrict ourselves to non-coherent receveived power to focus on the machine learning challenges (3D equivariance) that wireless channel modeling poses. We agree that, depending on the downstream use case of channel models, further channel information is required---and it is straightforward to embed with our tokenizer and process with our Wi-GATr backbone.
> Nonetheless, learning large-scale characteristics such as received power is an important open-problem and an active area of research [35, 29, 24].
>
> **Single antenna, does not cover larger bandwidths ... non-linearities can occur with larger bandwidths**
> Our setting is reflective of infinite bandwidth, since we sum up powers of individual physical paths (see line 245).
> Our model is trained directly on the received power and is agnostic to system-level details (e.g., bandwidth).
>
> We thank the reviewer again for their detailed feedback. We hope we were able to address their concerns and look forward to discussing more.

---

> > ### Comment · Reviewer_3vGS · 2024-08-14
> >
> > I would like to thank the authors for their response, which did address some of my concerns. I did appreciate the additional results which demonstrated the benefits of the tokenization scheme. Based on the response and update, I am going to keep my initial score. Thank you and all the best.

---

### Official Review · Reviewer_LEve · 2024-07-14

**Soundness:** 2
**Presentation:** 3
**Contribution:** 3
**Rating:** 4
**Confidence:** 3

**Summary:**

The paper presents the Wireless Geometric Algebra Transformer (Wi-GATr), a new architecture for simulating wireless signal propagation in 3D environments. This model utilizes geometric algebra to handle the geometric complexities of wireless scenes and ensures E(3) equivariance to respect the symmetries of the physical problem. The authors introduce two datasets, Wi3R and WiPTR, to benchmark their model. Wi-GATr outperforms existing baselines in terms of prediction fidelity and data efficiency, and it can solve both forward (signal prediction) and inverse (receiver localization and geometry reconstruction) problems in wireless communication.

**Strengths:**

1. The integration of geometric algebra for handling complex 3D geometric data and ensuring E(3) equivariance is a novel and effective approach. This addresses the core challenge of accurately modeling wireless signal propagation in diverse environments.

2. The paper provides a thorough evaluation of Wi-GATr against multiple baselines across various tasks, demonstrating superior performance in signal prediction, receiver localization, and geometry reconstruction.

3. Wi-GATr shows remarkable data efficiency, achieving high-fidelity predictions with significantly less training data compared to other models. This is particularly beneficial for scenarios where obtaining large amounts of training data is challenging.

4. The model's ability to handle both forward (predictive modeling) and inverse (localization and reconstruction) problems showcases its versatility and potential for a wide range of applications in wireless communication.

**Weaknesses:**

1. Limited Real-World Testing: While the model performs well on the introduced datasets, its application in real-world, dynamic environments remains underexplored. Additional experiments in more varied and complex real-world scenarios, such as urban or industrial settings, would strengthen the paper.

2. Scalability and Computational Load: The paper could provide more detailed insights into the computational requirements and scalability of Wi-GATr. Understanding the model's performance with larger datasets and more complex environments would be valuable for practical deployment.

3. Generalizability Across Frequencies: The model is tested at a specific frequency (3.5 GHz). Evaluating its performance across different frequencies and under various signal conditions would provide a more comprehensive understanding of its robustness and generalizability.


4. Detailed Case Studies: While the paper presents strong experimental results, including more detailed case studies or examples of practical applications, such as network design or optimization in real-world environments, would illustrate the model's impact and practical benefits.

**Questions:**

1. The paper focuses on indoor scenes for evaluation. Have you considered testing Wi-GATr on datasets that include outdoor or mixed environments to assess its robustness and generalizability?

2. The paper mentions solving inverse problems like receiver localization and geometry reconstruction. Can you provide more detailed examples or case studies where these capabilities significantly outperform traditional methods?

3. Given the computational complexity, how does Wi-GATr perform in real-time applications? Can it be integrated into real-time wireless network management systems, and if so, what are the challenges?

4. The datasets introduced are focused on indoor environments. Have you considered incorporating additional data sources, such as satellite imagery or LIDAR data, to enhance the model's accuracy and applicability in different environments?

5. How does the model adapt to changes in the environment, such as new constructions or changes in material properties? Is there a mechanism for updating the model dynamically to reflect these changes?

**Limitations:**

The authors briefly discussed the limitation in Sec. 6.

---

> ### Author Rebuttal · Authors · 2024-08-07
>
> We thank reviewer LEve for providing a detailed review of our paper.
> We are glad that the reviewer appreciates our equivariant approach to model diverse environments, the strong performance in our thorough evaluation, and the data efficiency and versatility of our approach on forward and inverse problems.
> Further, we appreciate the useful comments to improve our manuscript.
>
> Now, we address reviewer LEve’s concerns and also indicate if it is a common concern shared by fellow reviewers.
>
> **Limited real-world testing** (common concern - reviewers LEVe, 2AZ5)
> Firstly, we remark that evaluations on simulated data have their own merits: it allows evaluation on a large-scale (e.g., diverse environments, locations of transmitter/receiver) in a controlled setting and enables better analysis of approaches and results.
> While we agree with reviewers that it would be ideal to extend our evaluation to real-world datasets, we are limited by their availability.
> To the best of our knowledge, no relevant large-scale real-world dataset exist for the problem we tackle and hence previous works [29, 41, 35, 25] have also largely relied on simulated datasets.
> In fact, even the simulated datasets that exist (e.g., Wi3Rooms [41], Etoile/Munich [29]) are small-scale with low diversity in terms of 3D environments.
> This lack of high-quality wireless datasets with diverse scenes motivated us to generate and release two new datasets with this paper.
> Our symmetry considerations apply in the real-world and in simulation, thus our key message - that wireless channel modelling is a geometric problem and one should use symmetry-aware architectures and algorithms to solve it - should hold both in simulation and real-world measurements.
>
> **Scalability and Computational Load** (common concern - reviewers LEve, 6Syy)
> In Fig. 7 in Appendix D.1, we show timing measurements for a single room. Predicting the received power with Wi-GATr takes 0.02 s per Tx-Rx link, which is substantially faster than the ray tracers with similar accuracy.
>
> Thanks for the suggestion to study scalability to larger scenes, which we add in Fig. R4 in the rebuttal result page. Wi-GATr remains at sub-second latency for up to 10k tokens. The quadratic compute scaling of the attention mechanism with the number of tokens means that larger scenes with millions of mesh faces take hours to evaluate.
>
> **Generalizability Across Frequencies**
> Evaluating generalization across frequencies makes for an interesting experiment.
> However, the focus of this paper was generalization across *scenes*, since we believe this plays a larger role on wireless characteristics (unlike carrier frequencies, which are typically regulated and fixed).
> Nonetheless, our approach does not make any assumption on the carrier frequency and would make an ideal candidate for this experiment. We will consider this in future.
>
> **The paper focuses only on indoor evaluation ... considered outdoor/mixed?** (common concern - reviewers LEve, 2AZ5)
> Evaluation on outdoor scenarios would make a great addition.
> The physics (e.g., diffraction, reflections) of wireless signals would largely remain the same as a function of the 3D geometry in both indoor or outdoor settings.
> Nonetheless, in this paper, we choose indoor scenarios since we believe it is a more challenging scenario to evaluate influence of 3D scenes towards wireless characteristics.
> For instance, heights of surfaces play a bigger role (since signals can reflect off floors and ceilings) and transmissions through surfaces have significant contributions towards receive powers in non-LOS regions.
>
> **Comparison of inverse problems with traditional methods**
> We have provided an additional comparison to differentiable ray tracing (see *Sionna* in rebuttal pdf Fig. R2) on the forward problem and conclude it is not applicable to the inverse problem given its current accuracy. Other neural network methods we are aware of require data from the same scene. We evaluate on novel scenes not seen during training. We appreciate pointers to relevant methods that apply to this generalization setting.
>
> **How does Wi-GATr perform in real-time applications? ... Can it be integrated into real-time wireless network management systems ... what are the challenges?**
> The answer depends on the application and the required latency.
> Our approach predicts at sub-second latency for scenes with up to 10k mesh faces, see Fig. R2 and R4 in the rebuttal pdf.
> However, certain applications will require faster inference speeds.
> For instance, receiver localization through gradient descent might be too slow for real-time applications.
> Nonetheless, we expect that the inference speed of our architecture can still be optimized using standard tricks (e.g., compilation of computation graph, distillation, efficient diffusion samplers).
>
> **Have you considered additional data sources ... satellite imagery, LIDAR ... to enhance the model's accuracy and applicability**
> This is an interesting idea and would further highlight the versatility of learning-based simulation approaches. We have not considered it in this paper.
>
> **How does the model adapt to changes in the environment ... such as new constructions**
> Our approach takes the description of the environment as input, and our predictions adapt naturally to changing conditions.
> For instance, one of our out-of-distribution test scenarios "OOD layout" considers removing walls (see Table 2 of main paper, WiPTR dataset) and we observe Wi-GATr adapts.
> Similarly, our network predictions will change if the materials are changed.
>
> **Detailed case studies ... practical applications**
> We have shown the application of model to receiver localization in our extensive evaluation. This problem has practical use cases in robotics or, in the form of transmitter localization, in base station placement.
>
> We thank the reviewer again for their feedback. We hope we were able to address their questions and look forward to discussing further.

---

> > ### Comment · Reviewer_LEve · 2024-08-12
> > **Thanks for your response.**
> >
> > The unclear effects in real-world scenarios and outdoor environments reduce the contribution of this paper. The authors claim that indoor rooms are more critical and challenging. However, this does not mean that evaluating their work in outdoor scenarios is unnecessary. I acknowledge this work's novelty, and it's ok for me that the work is not evaluated on real-world data, but I think the experiments on the simulation data are still insufficient.

---

> > > ### Author Response · Authors · 2024-08-12
> > >
> > > We thank the reviewer for taking the time to read our rebuttal. We would like to clarify one thing: we did not mean to imply that outdoor evaluations are unnecessary or that indoor measurements are more critical – on the contrary, we would find an analysis on outdoor data just as valuable.
> > >
> > > However, we want to stress that the symmetries of the underlying physics are the same in indoor and outdoor problems. We therefore believe that our high-level findings about the benefits of geometric, symmetry-aware machine learning models should also apply to outdoor scenes.

---

### Author Rebuttal · Authors · 2024-08-07

We thank the reviewers for their thorough and constructive feedback.

We are glad to hear that reviewers **LEve**, **3vGS**, **6Syy**, and **yJCq** appreciate the versatility of our approach to forward, inverse, and generative problems. Reviewer **LEve** specifically highlights the data efficiency of our equivariant approach as well as our thorough evaluation, while reviewers **3vGS**, **2AZ5**, and **6Syy** appreciated our novel geometric tokenization scheme for wireless scenes.

The reviewers commented on and asked in-depth technical questions: about the quality of our wireless simulations, the relevance of non-coherent received power, whether our evaluation is relevant for real-world situations, and if we included all appropriate baselines. These points are important, and we address them in individual responses as well as at the end of this response.

We want to emphasize that we view our **main message** as partially orthogonal to the reviewers' concerns.
We introduce an *equivariant* model for wireless simulation.
Such an equivariant model considers the full 3D geometry of the scene and the symmetries of the underlying physics.
Although not shown explicitly, we argue that these symmetries hold for other channel characteristics as well.
We show that including inductive bias in the model leads to much better generalization to novel scenes than vanilla transformers.
We would appreciate hearing the thoughts of reviewers **3vGS**, **2AZ5**, and **6Syy** about our geometry- and symmetry-focused approach to wireless channel modelling.

&nbsp;

Let us now comment on specific technical **criticisms and questions**, which are immensely helpful for us in improving the paper.

**Limited real-world evaluation ... benchmarks with real-world traces missing** (reviewers LEVe, 2AZ5)
Firstly, we remark that evaluations on simulated data have their own merits: it allows evaluation on a large-scale (e.g., diverse environments, locations of transmitter/receiver) in a controlled setting and enables better analysis of approaches and results.
While we agree with reviewers that it would be ideal to extend our evaluation to real-world datasets, we are limited by their availability.
To the best of our knowledge, no relevant large-scale real-world dataset exist for the problem we tackle and hence previous works [29, 41, 35, 25] have also largely relied on simulated datasets.
In fact, even the simulated datasets that exist (e.g., Wi3Rooms [41], Etoile/Munich [29]) are small-scale with low diversity in terms of 3D environments.
This lack of high-quality wireless datasets with diverse scenes motivated us to generate and release two new datasets with this paper.
Our symmetry considerations apply in the real-world and in simulation, thus our key message - that wireless channel modelling is a geometric problem and one should use symmetry-aware architectures and algorithms to solve it - should hold both in simulation and real-world measurements.

**Scalability to large datasets and scenes .. complex meshes ... computation load** (reviewers LEve, 6Syy)
We already evaluate our approach on "large-scale" (reviewers 3vGS, yJCq) datasets (e.g., 1K WiPTR diverse scenes proposed in this paper) -- this is significantly larger than previous efforts.
In addition, we ran scalability experiments and found promising results.
As shown in Fig. R4 in the rebuttal pdf, Wi-GATr remains at sub-second latency until $\sim$10K mesh faces.
Although scenes with 500K mesh faces takes up to an hour with our vanilla implementation, based on preliminary experiments, this is faster than a conventional ray tracer.

**Baselines ... ablation ... benefits of tokenization unclear** (reviewers 3vGS, 6Syy)
While our new geometric tokenization scheme for wireless scenes is one of our key contributions, we did not carefully ablate its relevance empirically. We are grateful to the reviewers for pointing this out and added an ablation. In Fig. R1 of the additional result page, we now also show a naive transformer model that does not use our tokenization scheme. It performs substantially worse than both the transformer with our tokenization and our main method Wi-GATr.

**Baselines ... comparison with neural and differentiable ray tracers ... Sionna-RT [29]** (reviewers 6Syy, yJCq)
We appreciate the suggestion. We added a comparison to Sionna RT, in Figures R2 and R3 of the rebuttal result page. Since Sionna RT does not model transmission effects, we observe it gives poor results on our indoor scenes.

&nbsp;

We hope we were able to address the reviewers' concerns and clarify some misunderstandings. We are looking forward to an insightful discussion.

---

### Decision · Program_Chairs · 2024-09-25

**Decision:**

Reject

**Comment:**

The reviewers have indicated that the diffusion generator approach to modeling multi-path channels based on geometry and tokens is an interesting and potentially useful research direction.  Applying a group equivariant convolutional NN for this classical inherently linear reciprocal channel ray modeling application is logical.  The paper overall seems preliminary, and some key issues and questions have been raised by the reviewers.

There are a number of important issues raised such as scalability, use in a realistic scenario, claims of much lower complexity compared to ray tracing, only exploring power prediction, and limitations of experiments and comparisons.

Simulations are carried out for 3D indoor models at 3.5 GHz (Cellular 5G uses this frequency), where the wavelength is 86mm. Consequently, even very small unmodeled objects and perturbations in the environment or antenna position can cause significant channel state fluctuation. Note also that ray tracing models are only appropriate for relatively higher frequencies, and not for all wireless communications bands.  It may be useful to indicate the frequency bands of interest and discuss applicability, scalability, 3D modeling, and other issues in this context.

The results do not consider changes in the environment, e.g., doors, people. This impacts the forward and inverse modeling accuracy and statistical variation observed.  Similarly, small calibration errors in antenna response (and across multiple antennas), placement, and timing can also lead to significant channel changes.

Mesh-based methods have well known tradeoffs in mesh size versus modeling accuracy, that are not fully considered.  This impacts scalability of mesh methods and is relative to the wavelength of interest and the environmental complexity and extent.

The pending release of the indoor floor scenes dataset could be a useful contribution to the community.